# Emerging Contaminants: An Overview of Recent Trends for Their Treatment and Management Using Light-Driven Processes

**Brandon Chuan Yee Lee** †, **Fang Yee Lim** †, **Wei Hao Loh** †, **Say Leong Ong** and **Jiangyong Hu** *

Department of Civil and Environmental Engineering, Faculty of Engineering, National University of Singapore, Block E1A, #07-01, 1 Engineering Drive 2, Singapore 117576, Singapore; ceeblcy@nus.edu.sg (B.C.Y.L.); rlimtony@gmail.com (F.Y.L.); weihao.loh@u.nus.edu (W.H.L.); ceeongsl@nus.edu.sg (S.L.O.)
* Correspondence: ceehujy@nus.edu.sg
† These authors contributed equally to this work.

**Abstract:** The management of contaminants of emerging concern (CECs) in water bodies is particularly challenging due to the difficulty in detection and their recalcitrant degradation by conventional means. In this review, CECs are characterized to give insights into the potential degradation performance of similar compounds. A two-pronged approach was then proposed for the overall management of CECs. Light-driven oxidation processes, namely photo/Fenton, photocatalysis, photolysis, UV/Ozone were discussed. Advances to overcome current limitations in these light-driven processes were proposed, focusing on recent trends and innovations. Light-based detection methodology was also discussed for the management of CECs. Lastly, a cost–benefit analysis on various light-based processes was conducted to access the suitability for CECs degradation. It was found that the UV/Ozone process might not be suitable due to the complication with pH adjustments and limited light wavelength. It was found that $E_{EO}$ values were in this sequence: UV only > UV/combination > photocatalyst > UV/O$_3$ > UV/Fenton > solar/Fenton. The solar/Fenton process has the least computed $E_{EO} < 5$ kWh m$^{-3}$ and great potential for further development. Newer innovations such as solar/catalyst can also be explored with potentially lower $E_{EO}$ values.

**Keywords:** emerging contaminants; advanced oxidation process; environmental pollutant management; light-driven technology; $E_{EO}$

## 1. Introduction

Due to growing concern about the potential harm to the aquatic ecosystem and human health, a growing emphasis has been placed on the research of cost-effective treatment technology for contaminants of emerging concern (CECs). CECs are currently unregulated in most countries, with their treatment options not well documented [1]. The global prevalence of CECs in surface waters, groundwaters, and municipal wastewaters has resulted in a tightening of standards for emerging contaminants [2]. Despite guidelines set by local authorities, the occurrence of unregulated discharge often occurs due to a lack of strong legislation and ecotoxicity data on CECs [3]. CECs are resilient to the conventional biological treatment processes or have slow kinetics for biodegradation [4]. Uncontrolled release of CECs into water environments causes problems like environmental persistence, ecotoxicity, and potential harm to human health. Current detection methods of CECs also suffer from few limitations, such as prohibitive cost, time-consuming detection means, and low throughput. Rapid industry changes also make the treatment and regulation of CECs particularly challenging. For instance, Ahearn et al. [5] found that the replaced substance (GenX), for the recently regulated perfluorooctanoic acid (PFOA), had a more detrimental environmental impact. Hence, there is a pressing need for the treatment and management of CECs to keep up with global trends.

Light-driven advanced oxidation processes (AOPs) are a viable solution for the overall management of CECs, due to their high efficiency in degrading a wide spectrum of organics. The drastic decrease in the cost of UV-light emitting diode (LED) with a single peak emission wavelength has also allowed for more precise and cheaper UV-based detection of compounds [6]. Furthermore, the occurrence of CECs in secondary effluents with characteristically low turbidity and high UV transmittance >35–65% [7], makes light-based treatment particularly attractive.

Despite the multiple advantages of light-based systems, there is a lack of review for the light-based treatment and management of CECs. Hence, this review aims to provide an updated overview of the occurrence, treatment and management of CECs, using light-based technology. This review discusses a two-pronged approach for the treatment and management of CECs, highlighting recent applications within the last 3 years. Lastly, a cost–benefit analysis is done on the various light-based technology for greater insights into their potential development. Future directions for the light-driven process have also been suggested.

### 1.1. Global Occurrence of Contaminants of Emerging Concern (CECs) and Their Categorization

The global occurrence of CECs in developed and developing countries shows a pressing need to address this class of contaminants. A compilation of global occurrence of CECs detected in the environment, and their use and chemical structure is presented in Table 1.

**Table 1.** Global occurrence of CECs and their use in the industry.

| Category of Contaminant | Chemical Structure within the Contaminant | Name of Contaminant | Uses in Industry | Concentration Detected | References |
|---|---|---|---|---|---|
| Pharmaceuticals | Azetidine, Benzene | Amoxicillin | Antibiotics | Queensland: 6.9 µg/L<br>Delhi: Up to 172.6 ng/L<br>Ghana: Up to 0.0027 ng/L | [8–10] |
| | Halogenic-Benzene | Diclofenac | Anti-inflammatory drug | Algiers: 85.2 ± 9.3 ng/L<br>Saudi Arabian coastal waters: 10,221 ng/L<br>Lahore: 260–470 ng/L<br>WWTP effleunt from South Africa: 5.56–243.6 ng/L | [11–13] |
| | Benzene | Ibuprofen | Painkiller | Madrid: 4.1 ng/L<br>Algiers: 372.8 ± 19.8 ng/L<br>Saudi Arabian coastal waters: 127–660 ng/L<br>Lahore: 1728–2300 ng/L<br>Sea water Durban, South Africa: <0.17 ng/L | [11–15] |
| | Benzene, Piperidine | Acetaminophen (ACE) | Painkiller | Saudi Arabian coastal waters: 1234 and 2346 ng/L<br>Lahore: 12,120–13,880 ng/L | [11,12] |
| | Benzene, Pyrazine | Sulfamethoxazole (SMX) | Antibiotics | Mekong Delta: 21 ng/L<br>Jiangsu Province: 63.6 ng/L<br>Madrid: 162–530 ng/L<br>Ghana: 0.013–2.861 ng/L | [10,14,16, 17] |
| | Benzene, 7-member ring | Carbamazepine | Anticonvulsant | Various plants in the USA: 2–207 ng/L<br>Spain treatment plants: <54 ng/L<br>Hartbeespoort Dam catchment and the uMngeni River estuary: up to 94 ng/L | [18–20] |

Table 1. *Cont.*

| Category of Contaminant | Chemical Structure within the Contaminant | Name of Contaminant | Uses in Industry | Concentration Detected | References |
|---|---|---|---|---|---|
| Food additive | Oxadiazine | Acesulfame potassium (ACE-K): | Artificial sweetener in food and beverages | Jiangsu: 2.9 μg/L to 0.20 mg/L German Elbe river: 100 to 900 mg/s (mass load) | [21,22] |
| | Benzothiazole | Sucralose | | Jiangsu: up to 3.6 μg/L. | [21] |
| Pesticides | Triazine | Atrazine | Herbicide | Jiaozhou Bay: 76 ng/L Ctalamochita river basin: 0.23 to 0.26 ng/L (urban), 0.28 to 3 ng/L (rural) Hartbeespoort Dam catchment (South Africa): up to 1570 ng/L | [23–25] |
| Industrial chemicals | Dioxane | 1,4-Dioxane | Organic solvents | Sant Joan Despí: 4360 (average), 32,370 (max) Oder River:143–2245 ng/L Rhine/Main River:110–850 ng/L | [26,27] |
| | Pyrazole, 6 membered heterocyclic ring | Caffeine | Food and beverage industry | Saudi Arabian coastal waters: 7708 ng/L Various plants in the USA: 7–687 ng/L Madrid: 5010–65 625 ng/L WWTP effluent from South Africa: 85.76–4878 ng/L | [12,14,18] |
| | - | Perfluorooctane sulfonate (PFOS) and Perfluorooctanoic acid (PFOA) | Industrial manufacturing and use and disposal of PFAS-containing products, | Worldwide: 0.2–1630.2 ng/L Singapore: 532–1060 ng/L (WWTP treated effluent) WWTP effluent from Kampala, Uganda: PFOS (1.3–1.5 ng/L) and PFOA (1.5–2.4 ng/L) | [28–30] |
| | - | N-Nitroso-dimethylamine (NDMA) | A by-product of industrial processes that use nitrates and/or nitrites. | Various plants in the USA: 12–321 ng/L | [18] |
| | Phenols | Bisphenol A (BPA) | Plastic formation | Yamuna/Cooum River: 1420–14,800 ng/L Zhujiang/Dongjiang River: 101–2310 ng/L Zhujiang/Dongjiang WWTP: 29,400 ng/L Riyadhm Saudi Arabia/Drinking water: 291–41,190 ng/L | [31–33] |
| Personal care products (PCPs) | Benzene | Diethyltoluamide (DEET) | Insect repellent | Arizona: 1570–15,200 ng/L | [34] |
| | Benzene, 5- member cycloalkane | Galaxolide | Synthetic musk | Madrid: <24 971 ng/L Lubbock: 3789–10,525 ng/L | [14,35] |

**Table 1.** *Cont.*

| Category of Contaminant | Chemical Structure within the Contaminant | Name of Contaminant | Uses in Industry | Concentration Detected | References |
|---|---|---|---|---|---|
| **Disinfection by-products (DBPs)** | Dihalobenzoquinones | 2,6-dichloro-1,4-benzoquinone | Disinfection by-product from water treatment | Canada WWTP: 165.1 ng/L China Drinking WTP: 2.6–19.70 ng/L | [36,37] |
| | Iodotrihalomethanes | Dibromoiodomethane (DBIM), Chlorodi-iodomethane (CDIM), Bromodiiodomethane (BDIM), Iodoform (TIM) | Disinfection by-product from water treatment | China Drinking WTPs: 0.007–0.23 ng/L Australia Advanced water recycling plant: <1–7 ng/L | [38,39] |

As seen from Table 1, the global prevalence of CECs in water bodies calls for better management strategies and treatment technology. Management of CECs could be done by first categorizing CECs. Categorization of CECs based on the usage and chemical structure would provide insights into the fates and potential degradation performance of newer CECs. Concerning treatment by light-driven processes, CECs should be best categorized based on their chemical structures. Polarity and hydrophobicity ($K_{ow}$) of CECs are dependent on their molecular structures and hence CECs with similar chemical structures, generally exhibit similar chemical characteristics, removal mechanisms, spectral properties and toxicity [40]. CECs with similar structures were reported to have similar photodegradation performance an illustrated example is shown in the Supplementary S1. Structures such as halogens present in CECs result in a resonant stabilized structure [41]. Whereas chemical structures like electron-donating hydroxyl (–OH) and amino (–NH$_2$) functional groups affected the resonance stability of ring structures, resulting in different degradation performances [42]. Chemical stability and optical properties are directly correlated to CECs degradation performance by light-based processes.

*1.2. Conventional Detection of CECs in Water Bodies*

Conventionally, CECs are concentrated via various extraction methods to improve their detection accuracy, such as liquid–liquid extraction (LLE), solid-phase extraction (SPE), solid-phase microextraction (SPME), stir bar sorptive extraction (SBSE), matrix solid-phase dispersion (MSPD), pressurized liquid extraction (PLE), ultrasound-assisted extraction (UAE), microwave-assisted extraction (MAE), and liquid-phase microextraction [39,43]. A conventional mode of detection of CECs is usually performed using liquid chromatograph (LC) and gas chromatograph (GC) methods [44,45]. The key difference between the two methods is the mobile phase used. GC uses an inert gas (like helium) for its mobile phase, while LC uses a polar solvent (like water or methanol) for its mobile phase.

GC requires the volatilization of CECs and is detected via various means. GC coupled with electron capture detection (ECD) has been widely applied to analyze chlorinated-DBPs from chlorina (mina) ted water [46]. ECD is highly sensitive to electronegative compounds, such as compounds containing halogens, nitrogen, and sulfur. Andersson et al. [47] demonstrated GC equipped with a halogen-specific detector (XSD) for the simultaneous determination of traditional and emerging halogenated-DBPs. XSD shows high selectivity towards halogenated compounds only. It is noted that the methods proposed above are highly specific to their structure type and hence the classification of CECs (in particularly newer CECs) based on their chemical structures may be an attractive means to assess the suitability of present detection methods. GC can also be coupled with a quadrupole time-of-flight (QTOF)-MS for the detection of micropollutants [44]. More conventionally, GC is coupled with a mass spectrophotometer (MS) for the detection of CECs.

However, GC methods might not be suitable for compounds that are susceptible to thermal decomposition during acquisitions [45]. Liquid chromatograph (LC) uses the principle of mass transfer in a polar liquid and does not require sample volatilization, hence preventing compound degradation and the formation of new products under high heat conditions [45]. Liquid chromatograph (LC) coupled with triple quadrupole (QqQ) or quadrupole linear ion trap (QqLIT) MS/MS is one of the most widely used for CECs quantification [14], whereas LC-(QTOF)-MS uses the principle of molecular weight and has been used to study transformation products and the reaction pathway of PPCPs and the formation of DBPs [48]. Detection and characterization of DBPs in complex matrices can be performed by Fourier transform ion cyclotron resonance mass spectrometry (FT-ICR MS) [48].

It is important to note that the use of the various methods is highly specific to the physico-chemical property of the CECs, with some CECs more suitable for a certain type of detection method [44]. The definition of CECs by their similar chemical structure mentioned in earlier segments is hence an attractive means not only to categorize the CECs but also to know its suitable detection methods. However, conventional methods of testing are labor-intensive and have limited application for online detection due to the expensive equipment needed [49].

## 2. Overview of Light-Driven Processes

Due to the global occurrence of CECs in various waters and the difficulty in detection of these compounds, there is a pressing need for the overall treatment and detection of CECs. Herein, a two-pronged approach is proposed for the treatment and management of CECs (Figure 1). Light-based treatment can be categorized into UV/oxidant, UV/ozone, photo-Fenton, and photocatalysis, with each having a range of UV wavelength and factors affecting its operation, which is discussed below, whereas the detection methods discussed involve rapid and alternative methods for the detection of CECs in wastewater.

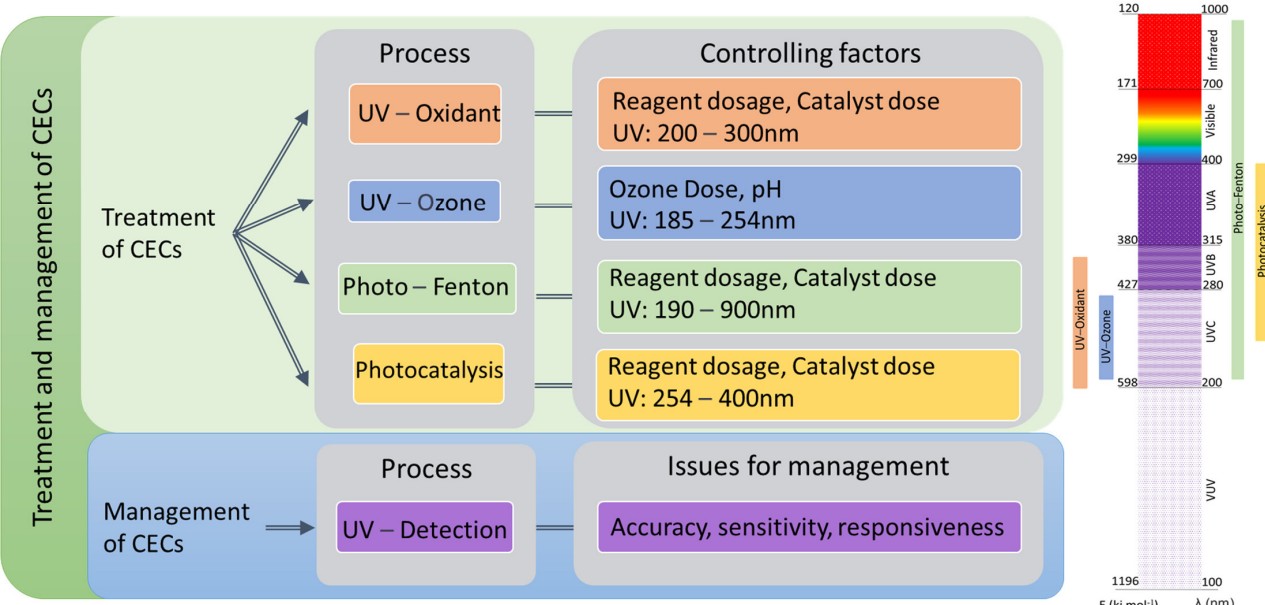

**Figure 1.** Overview of the treatment and management of CECs.

### 2.1. Mechanism of Light-Driven Processes

Light-driven processes involve two main reactions: (1) direct photolysis and (2) generation of highly reactive oxidative substrates, such as hydroxyl ($\bullet$OH), chlorine ($\bullet$Cl), sulfate ($SO_4\bullet^-$), and hydroperoxyl radicals ($HO_2\bullet$), by catalytically converting water or oxidants for the degradation of wastewater. Direct photolysis occurs when the light energy used ($E_\lambda$) is more than the associated bond energy of the contaminants [50]. The energy

supplied by the light processes could be approximated to a specific wavelength, as shown in Supplementary S2, whereas radicals react directly with CECs and degrade them. The generation of reactive species is dependent on the process and will be further elaborated below. The key reaction mechanisms and graphical illustration of the mechanisms are summarised in Table 2 and Figure 2.

**Table 2.** Mechanism of conventional light-driven AOPs.

| Name of AOP | Type | Mechanism | References |
|---|---|---|---|
| UV/Oxidant | UV-$H_2O_2$ | $H_2O_2 + hv \rightarrow 2HO^\bullet$ | [51] |
| | UV-persulfate | $S_2O_2^{2-} + hv \rightarrow 2SO_4^{\bullet-}$ <br> $SO_4^{\bullet-} + H_2O \rightarrow SO_4^{2-} + HO^\bullet + H^+$ <br> $SO_4^{\bullet-} + OH^- \rightarrow SO_4^{2-} + HO^\bullet$ | [52] |
| | UV-chlorine | $HOCl \rightarrow H^+ + OCl^-$ (pK$_a$ = 7.5 at 25 °C) <br> $HOCl + hv \rightarrow HO^\bullet + Cl^\bullet$ <br> $OCl^- + hv \rightarrow O^{\bullet-} + Cl^\bullet$ <br> $O^{\bullet-} + H_2O \rightarrow HO^\bullet + OH^-$ <br> $Cl^\bullet + OH^- \rightarrow ClHO^{\bullet-}$ | [53] |
| UV/Ozone | Microbubble | $O_3 + hv \rightarrow O_2 + O(^1D)$ <br> $O(^1D) + H_2O \rightarrow H_2O_2$ <br> $H_2O_2 + hv \rightarrow 2HO^\bullet$ <br> $3O_3 + hv + H_2O \rightarrow 2HO^\bullet + 4O_2$ | [54] |
| Photo-Fenton | Homogenous | $H_2O_2 + Fe^{2+} \rightarrow Fe^{3+} + HO^\bullet + OH^-$ <br> $H_2O_2 + Fe^{3+} \rightarrow Fe^{2+} + HO_2^\bullet + H^+$ <br> $hv + Fe^{3+} \rightarrow Fe^{2+} + HO^\bullet Fe^{3+} + HO_2^\bullet \rightarrow Fe^{2+} + O_2 + H^+$ | [55] |
| | Heterogenous | $\equiv Fe^{2+} + H_2O_2 \rightarrow \equiv Fe^{3+} + OH^- + HO^\bullet$ <br> $\equiv Fe^{3+} + H_2O_2 \rightarrow \equiv Fe(OOH)^{2+} + H^+$ <br> $\equiv Fe(OOH)^{2+} \rightarrow \equiv Fe^{2+} + HO_2^\bullet$ <br> $\equiv Fe^{2+} + hv \rightarrow \equiv Fe^{2+} + HO^\bullet$ <br> $[Fe^{3+}L] + hv \rightarrow [Fe^{3+}L]^* \rightarrow Fe^{2+} + L^\bullet$ | [55,56] |
| Photocatalysis | $TiO_2$ | $TiO_2 + hv \rightarrow e_{cb}^- + h_{vb}^+$ <br> $h_{vb}^+ + OH \rightarrow HO^\bullet$ <br> $h_{vb}^+ + H_2O \rightarrow HO^\bullet + H^+$ <br> $e_{cb}^- + h_{vb}^+ \rightarrow heat$ | [57,58] |
| | ZnO | $ZnO + hv \rightarrow e_{cb}^- + h_{vb}^+$ <br> $e_{cb}^- + O_2 \rightarrow O_2^{\bullet-} h_{vb}^+ + OH \rightarrow HO^\bullet.$ | [59] |

For UV/oxidant processes, UV light directly interacts with added reagents to produce highly reactive radicals [59,60]. The wavelength of light required is between 200–300 nm. For the UV/Ozonation process, the UV light interacts with ozone to form hydrogen. The produced hydrogen peroxide then further reacts with UV to form hydroxyl radicals [54]. In the homogenous photo-Fenton process, the $Fe^{2+}$ in the aqueous form reacts directly with hydrogen peroxide to form hydroxyl radicals and $Fe^{3+}$. Ferrous ions ($Fe^{2+}$) are then photochemically regenerated by the photo-reduction of ferric ions ($Fe^{3+}$) [55]. The wavelength of light required is reported to be between 190–900 nm. In heterogenous photo-Fenton, iron can also partake in dissolution to form aqueous forms of iron species, which would subsequently follow a similar reaction mechanism of Ferrous ions in homogenous photo-Fenton [54,55]. This step is further illustrated in Figure 2c. It should also be noted that the mechanism for heterogenous photo-Fenton is largely dependant on the pH operation and the iron species used. The wavelength of light required is reported to be between 180–400 nm, whereas in photocatalysis, light irradiation of catalyst produces electron–hole pairs within the conduction and valence bands. The subsequent excitation and recombination of the energy state of the catalyst form reactive radicals [61]. This illustration

is further displayed in Figure 2d. The wavelength of light required is reported to be between 254–400 nm.

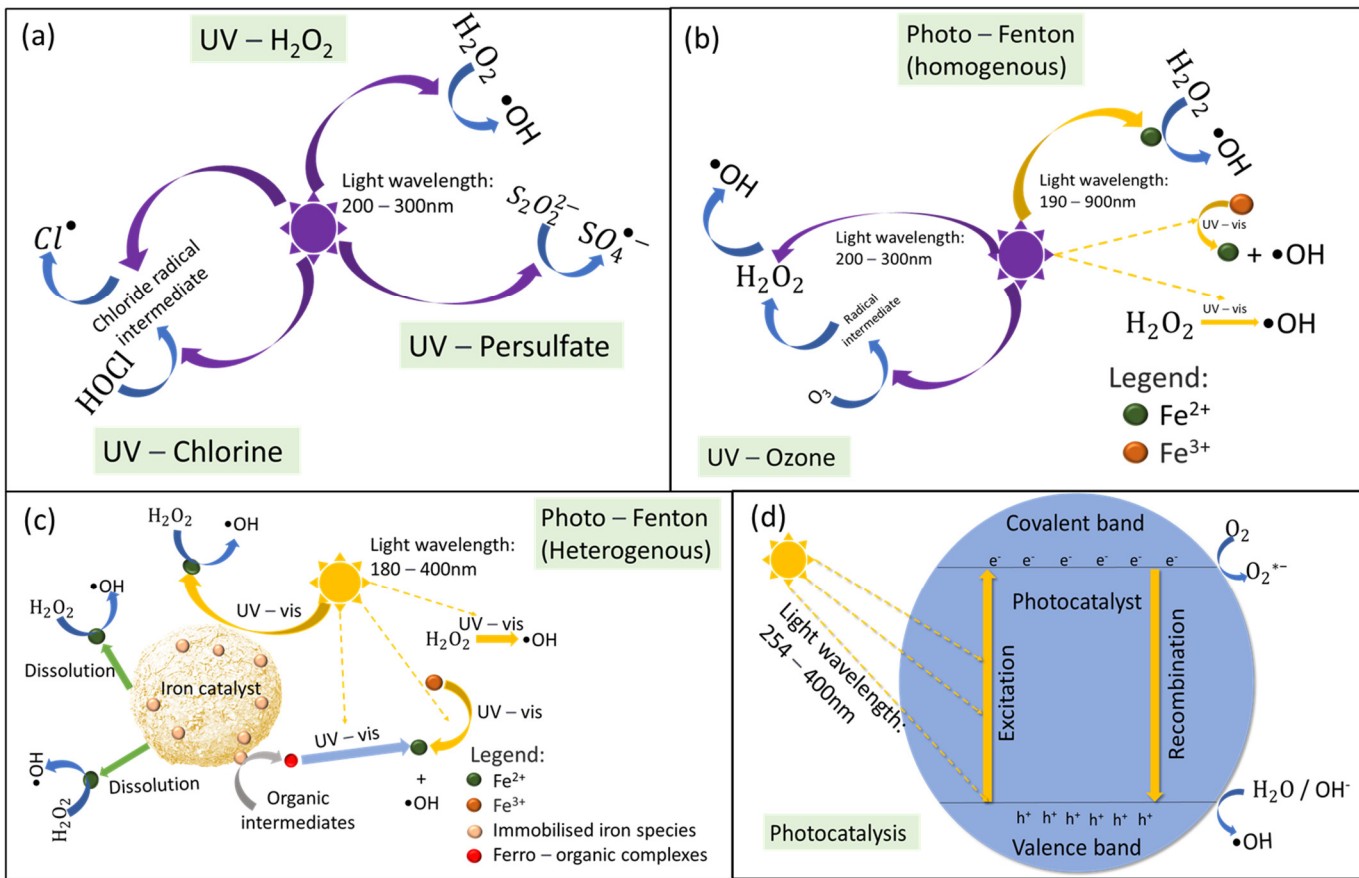

**Figure 2.** Graphical illustration of the reaction mechanisms for various light-driven processes: UV/oxidant (**a**), UV/ozone and photo-Fenton (homogenous) (**b**), photo-Fenton (heterogenous) (**c**), photocatalyst (**d**).

### 2.2. Critical Factors That Affect Light-Driven Processes

Generally, light-driven processes are affected by UV absorbance, pH, water matrix, reagent dosages, and UV sources (Table 3). The organic and inorganic composition of water affects the performance of light drive processes either by inhibition of chemical processes or inhibition of light penetration. The presence of turbid water or NOM reduces the photocatalytic effect due to charge carrier generation and UV absorbance [60,62]. The pH affects the process due to the inhibition effect from the stability of the oxidant or catalyst [63,64]. Too low a pH would affect the stability of metal-based catalyst in Fenton-based and photocatalyst systems [63–65]. Oxidant dosage and catalyst dosage affect the performance of CEC degradation since they affect the generation of oxidative species needed for CEC degradation. In excess, it might have scavenging effects or even change the physical characteristic of the water (i.e., light penetration), which affects the overall CEC degradation performance [66–68]. Lastly, the irradiation source also affects the CEC degradation performance of light-based systems. Generally, light waves with shorter wavelengths carry more energy and hence better performance [66–68]. However, a shorter wavelength of light is more energy-intensive and ongoing research is being done to have systems that use a longer wavelength of light or even natural sources of light for irradiation.

**Table 3.** Critical factors that affect light-driven processes.

| Factors that Affect the Process | Effects on Process | References |
|---|---|---|
| Low UV-absorbance | UV/Ozone:<br>- Reduces UV dose to generate $^\bullet$OH radicals<br>- Works well only with UVC rays, due to the highest molar adsorption coefficient<br><br>Photocatalysis:<br>- Affects UV dose for degradation<br><br>Photo-Fenton:<br>- Reduces UV dose to generate $^\bullet$OH radicals<br><br>Photocatalysis:<br>- Affects the charge carrier generation for photocatalytic activity and is only effective in selected wavelengths | [65,69] |
| Low pH | UV/Ozone:<br>- Ozone degradation rate<br>- Stability of pollutant<br><br>UV/Oxidant:<br>- Stability of peroxide ions<br>- Stability of $ClO^-$<br><br>Photo-Fenton:<br>- Stability of $Fe^{2+}$ ions and peroxide ions<br>- Leaching of iron species in low pH<br><br>Photocatalysis:<br>- Leaching of metal species in low pH<br>- Attraction between catalyst and pollutants | [63–65] |
| Water matrix: Presence of natural organic matter (NOM) | UV/Ozone:<br>- Adsorption of UV<br>- Scavenge $^\bullet$OH radicals<br><br>UV/Oxidant:<br>- UV-absorbing humic matter fraction easily degraded but non-absorbing low-molecular-weight (LMW) fractions get concentrated<br><br>Photo-Fenton:<br>- Ligand-to-metal charge transfer (LMCT) effect inhibiting Fenton reaction<br><br>Photocatalysis:<br>- Higher molecular weight fractions adsorbed onto catalysts and acted as electron-hole scavengers to reduce the photocatalytic degradation rate<br>- Scavenge $h^+$<br>- Occluding radicals generation sites | [60,62,70–73] |
| Water matrix: Presence of $O_2$ | Photocatalysis:<br>- Acts as a scavenger for $e^-$ to stop the recombination with $h^+$<br><br>Photo-Fenton:<br>- Oxygen competed against hydroxyl radical for degradation of organics | [59,74–77] |

**Table 3.** *Cont.*

| Factors that Affect the Process | Effects on Process | References |
|---|---|---|
| Excess oxidant/catalyst dosage | UV/Ozone:<br>- Increased $O_3$ dose, increase degradation rates, the excess might have scavenging effects<br>Photo-Fenton:<br>- Scavenge $^\bullet$OH radicals if $H_2O_2$ is in excess<br>- $H_2O_2$ needed to generate $^\bullet$OH radicals<br>- Reduce light penetration if added in excess<br>UV/Oxidant:<br>- Self-scavenge, reducing degradation rates<br>Photocatalysis:<br>- Reduce light penetration if added in excess | [59,78–80] |
| Type of irradiationof light used | Photo-Fenton:<br>- Solar/Fenton can achieve the same efficiency as UV/Fenton but at slower degradation rates<br>Photocatalysis:<br>- Doping of catalyst allows the bandgap to be reduced, increasing the capacity to utilize both visible light and UV | [66–68] |

*2.3. UV/Oxidant*

Due to the ease of operations, UV/oxidant processes have been widely applied in synthetic and spiked wastewater effluent (Table 4). UV/oxidant processes use a variety of different oxidative species (chlorine, peroxide, nitrates, sulfate, and its derivatives) for the generation of oxidative radicals. Generally, the reaction is operated in the neutral pH, with a hydraulic retention time (HRT) between 20–180 min and a vast majority of the systems use high-energy UVC irradiation systems to activate the oxidants. Most of the investigations using UV/Oxidant processes have focused on the removal of target compounds spiked in synthetic matrices at laboratory and pilot scales with a good degree of degradation of between 80–90% of the CECs.

**Table 4.** Current application of UV/oxidant processes.

| Category | Wastewater/Compounds | Operation Condition | Removal Efficiency | References |
|---|---|---|---|---|
| **UV/peroxide** | Wastewater treatment plant effluent | • UV/$H_2O_2$<br>• Low-pressure UV lamp<br>• UV wavelength: 254 nm (UVC)<br>• Power output: 2 mW/cm$^2$<br>• Chemical dose: 100 mg/L ($H_2O_2$)<br>• HRT: 60 min | The initial concentration of contaminant: ~400 ng/L<br>DOC: 70%<br>DON: 20–30%<br>NDMA: ~80% | [81] |
| | Wastewater treatment plant effluent | • UV/$H_2O_2$<br>• Medium-pressure UV lamp<br>• UV wavelength: 254 nm (UVC)<br>• Power output: 3 kW<br>• Chemical dose: 10 mg/L ($H_2O_2$)<br>• HRT: up to 60 min | Initial concentration of contaminant: 150–300 ng/L and 10–20 µg/L of bisphenol A (BPA).<br>perfluorooctanoic acid (PFOA): <30%<br>perfluorooctane sulfonate (PFOS): <30%<br>N-nitrosodimethylamine (NDMA): ~60%<br>Diclofenac: ~90%<br>(BPA): ~60%<br>2-Methylisoborneol (MIB): ~80%<br>Geosmin: ~90%<br>17 beta-estradiol (E2): ~70% | [82] |

**Table 4.** *Cont.*

| Category | Wastewater/ Compounds | Operation Condition | Removal Efficiency | References |
|---|---|---|---|---|
| | Wastewater treatment plant effluent doped CECS | • UV/H$_2$O$_2$<br>• Low-pressure UV lamp (LP-UV)<br>• UV wavelength: (UVC)<br>• Chemical dose: 100 mg/L<br>• Power: 5.3 W<br>• UV fluence: 55.2 mW/cm$^2$<br>• HRT: 80 min | Initial concentration of contaminant: 1 µM<br>Nonylphenol ethoxylated: 99% removal | [83] |
| | Synthetic wastewater, simulating pharmaceuticals discharges | • UV/H$_2$O$_2$<br>• Low-pressure UV lamp, UV-LED<br>• UV wavelength: 254 nm (UVC)<br>• Power output: 15 W<br>• Chemical dose: 1 mM (H$_2$O$_2$)<br>• HRT: 120 min<br>• UV Fluence: 0.1 mW/cm$^2$ (UV lamp) 0.13 mW/cm$^2$ (UV-LED) | Initial concentration of contaminant: 1 µM<br>Triclosan: ~80% (both systems)<br>Diclofenac: ~80% (both systems) | [84] |
| | RO treatment plant effluent | • UV/H$_2$O$_2$<br>• Low-pressure UV lamp (LP-UV) and UV-LED<br>• UV wavelength: 254 nm (UVC)<br>• Power output: 15 W<br>• Chemical dose: 10 mg/L (H$_2$O$_2$)<br>• HRT: 0.44 h<br>• UV fluence rate: 0.1 mW/cm$^2$ (LP-UV) 0.03 mW/cm$^2$ (UV-LED)<br>• pH: buffered at 7.4 | Initial concentration of contaminant: 1 µM<br>Ibuprofen (IBP): ~80%<br>Triclosan (TCS): ~80%<br>Estrone (E1): ~80%<br>Diclofenac: ~80% (mostly contributed by UV)<br>Bisphenol A (BPA): ~60% | [85] |
| | Sulfolane containing wastewater | • UV/H$_2$O$_2$<br>• UV wavelength: NA<br>• Chemical dose: 40–70 mg/L [H$_2$O$_2$]<br>• UV Power: 2140 W<br>• HRT: 10–30 min<br>• pH: 7.75–9.07 | Initial concentration of contaminant: 200 µg/L<br>Sulfolane removal: 36–89% | [86] |
| UV/peroxide and UV/chlorine | Wastewater treatment plant effluent | • UV/H$_2$O$_2$ and UV/Cl<br>• Medium-pressure UV lamp<br>• UV wavelength: 200–400 nm (UVA, UVB, UVC)<br>• Power output: 1 kW<br>• Energy consumption: 0.4 kW, 1 m$^3$/h<br>• Chemical dose: 6 mg/L (H$_2$O$_2$, Cl$_2$) | 4t-octylphenol: 65% (UV/Cl), 0% (UV/H$_2$O$_2$)<br>bisphenol A (BPA), 4-nonylphenols, TCPP: ~90% (UV/C, UV/H$_2$O$_2$)<br>Diclofenac: ~90% (UV/C, UV/H$_2$O$_2$)<br>DEET, HHCB: ~40% (UV/C, UV/H$_2$O$_2$) | [87] |
| | Wastewater treatment plant effluent | • UV/H$_2$O$_2$, Solar/H$_2$O$_2$, UV/Cl<br>• Medium-pressure UV lamp<br>• UV wavelength: 254 nm (UVC)<br>• Power output: 230 W<br>• sup>· Energy consumption: 87.7 W/m$^2$<br>• Chemical dose: 10 mg/L (Cl), 10 mg/L (H$_2$O$_2$)<br>• HRT: 60 min | Initial concentration of contaminant: 1:1 dilution with real wastewater 200 µg/L<br>Sulfamethoxazole: ~90% ((UV/Cl):3 min, (UV/H$_2$O$_2$): 6 min)<br>Imidacloprid: ~90% (UV/Cl):8.5 min, (UV/H$_2$O$_2$): 12 min<br>Carbamazepine: ~90% (UV/Cl): 6 min, (UV/H$_2$O$_2$): 25 min<br>Diclofenac: ~90% ((UV/Cl): 3 min, (UV/H$_2$O$_2$): 6 min<br>Blended CEC matrix: 82% (UV/Cl) and 87% (UV/H$_2$O$_2$) (60 min) | [88] |

**Table 4.** *Cont.*

| Category | Wastewater/ Compounds | Operation Condition | Removal Efficiency | References |
|---|---|---|---|---|
| UV/persulfate and UV/peroxide | Synthetic wastewater, simulating pharmaceuticals discharges | • sup>· UV/$H_2O_2$ and UV/$S_2O_8^{2-}$<br>• Low-pressure UV lamp (LP-UV) and UV-LED<br>• UV wavelength: 254 nm (UVC- using a UV filter)<br>• Power output: 280 W<br>• Chemical dose: Molar ratio of chemical dose: compound = 20:1<br>• HRT: 0.44 h<br>• pH: buffered at 7.4 | Initial concentration of contaminant: 2.2, 3.0, 5.2, and 5.5 μM (LP, FRSM, CAF, and CBZ)<br>losartan potassium (LP): 85% ((UV/$S_2O_8^{2-}$): 290 mJ cm$^{-2}$ (UV/$H_2O_2$): 620 mJ cm$^{-2}$)<br>furosemide (FRSM): 85% ((UV/$S_2O_8^{2-}$): 290 mJ cm$^{-2}$ (UV/$H_2O_2$): 620 mJ cm$^{-2}$)<br>caffeine (CAF): 85% ((UV/$S_2O_8^{2-}$): 290 mJ cm$^{-2}$ (UV/$H_2O_2$): 620 mJ cm$^{-2}$)<br>carbendazim (CBZ): 85% ((UV/$S_2O_8^{2-}$): 290 mJ cm$^{-2}$ (UV/$H_2O_2$): 620 mJ cm$^{-2}$) | [89] |
| | Wastewater treatment plant effluent spiked with synthetic compounds | • UV/$S_2O_8^{2-}$ and UV/$HSO_5^-$<br>• Low-pressure UV lamp (LP-UV)<br>• UV wavelength: 254 nm (UVC- using a UV filter)<br>• Mean intensity: 4.17 mW/cm$^2$<br>• Power output: 9 W<br>• Chemical dose: 1.0 mM<br>• HRT: 30 min<br>• pH: 7.8 ± 0.2 | Initial concentration of contaminant: 500 μg/L<br>Carbamazepine (CBZ): >90%<br>Crotamiton (CRMT): >90%<br>N,N-diethyl-meta-toluamide (DEET): >90% (80% UV/$S_2O_8^{2-}$)<br>Gemfibrozil (GEM): >90%<br>Ibuprofen (IBP): >90%<br>Trimethoprim (TMP): >90%<br>TOC removal: 31.8% (UV/$HSO_5^-$) and 33.7% (UV/$S_2O_8^{2-}$) | [90] |
| UV/persulfate | Synthetic wastewater, simulating pharmaceuticals discharges | • UV/P<br>• Low-pressure UV lamp (LP-UV)<br>• UV wavelength: 254 (UVC)<br>• Chemical dose: 1 mM PS<br>• Power: 0.44 μE/s<br>• HRT: 90 min<br>• pH: 6.5 | Initial concentration of contaminant: 32.8 μM<br>Methyl paraben: 98.9% removal | [91] |
| | Synthetic wastewater, simulating pharmaceuticals discharges | • UV/P<br>• Low-pressure lamps<br>• UV wavelength: 254 (UVC)<br>• Chemical dose: 0.25 mM PS<br>• Power: 4.9 W<br>• HRT: 100 min<br>• pH: 6.07 | Initial concentration of contaminant: 31 μM<br>Chloramphenicol: 100% removal | [92] |
| | Synthetic wastewater, simulating pharmaceuticals discharges | • UV/P<br>• Low-pressure lamp<br>• UV wavelength: 254 (UVC)<br>• Chemical dose: 0.2 mM PS<br>• Power: 10 W<br>• HRT: 180 min<br>• pH: 6 | Initial concentration of contaminant: 2 μM<br>Haloacetonitriles: 99.8% removal | [93] |
| | Synthetic wastewater, simulating pharmaceuticals discharges | • UV/P<br>• Medium-pressure lamp<br>• UV wavelength: 200–300 (UVC)<br>• Chemical dose: 1 mM PS<br>• Power: 2.8 kW<br>• pH: 5.85 | Initial concentration of contaminant: 23.69 μM<br>Sulfamethoxazole: 97% removal | [94] |
| | Synthetic wastewater, simulating pharmaceuticals discharges | • UV/P<br>• Low-pressure lamps<br>• UV wavelength: 254 (UVC)<br>• Chemical dose: 0.25 mM, PMS<br>• Power: 15 W<br>• HRT: 180<br>• pH: 5.8 | Initial concentration of contaminant: 3.43 μM<br>Lindane: ~90% removal | [95] |

**Table 4.** *Cont.*

| Category | Wastewater/ Compounds | Operation Condition | Removal Efficiency | References |
|---|---|---|---|---|
| | Wastewater treatment plant effluent and surface water spiked with synthetic compounds | • UV/$H_2O_2$<br>• Low-pressure UV lamp (LP-UV)<br>• UV wavelength: 254 nm (UVC)<br>• Power output: 15 W<br>• Chemical dose: 50 μM<br>• UV fluence: 4.23 mW/cm$^2$<br>• HRT: 30 min<br>• pH: natural pH of wastewater | Initial concentration of contaminant: 1 μM each<br>1H-benzotriazole (BZ): ~0% (secondary effluent), ~80% (surface water)<br>N,N-diethyl-m-toluamide (DEET): ~20% (secondary effluent), ~70% (surface water)<br>Chlorophene (CP): ~90% (secondary effluent), ~90% (surface water)<br>3-methylindole (ML): ~90% (secondary effluent), ~90% (surface water)<br>Nortriptyline hydrochloride (NH): ~90% (secondary effluent), ~90% (surface water) | [96] |
| | GENx | • UV/$S_2O_8{}^{2-}$ and UV/sulfite<br>• Low-pressure UV lamp (LP-UV)<br>• UV wavelength: 253.7 nm<br>• Power output: 35 W<br>• Chemical dose: 20 mM<br>• UV fluence: 8.0 mW/cm$^2$,<br>• HRT: 180 min<br>• pH: 10 | Initial concentration of contaminant: 1 mg/L<br>Perfluorooctanoic acid (PFOA): <90% (UV/sulfite), ~10% (UV/$S_2O_8{}^{2-}$)<br>Hexafluoropropylene oxide dimer acid (GenX): <95% (UV/sulfite), ~35% (UV/$S_2O_8{}^{2-}$) | [97] |
| **UV/nitrate** | Wastewater treatment plant effluent doped with CECS | • Solar/PAA UV/PAA<br>• Low-pressure UV lamp (LP-UV) and sunlight<br>• UV wavelength: 254 nm (UVC)<br>• Chemical dose: 10 mg/L<br>• Power: 230 W<br>• UV fluence: 40 kJ/L<br>• HRT: 30 min | Initial concentration of contaminant: 100 μg/L<br>Carbamazepine (CBZ): 30% (sunlight), ~70% (UVC)<br>Diclofenac (DCF): >90% (sunlight), >90% (UVC: 2 kJ L$^{-1}$)<br>Sulfamethoxazole (SMX): >90% (sunlight), ~100% (with UVC alone) | [98] |
| | Generic CECs | • UV/$NO_2{}^-$<br>• Low-pressure UV lamp (LP-UV)<br>• UV wavelength: 365 nm<br>• Chemical dose: 0.5 mg-N/L<br>• UV fluence: 3.05 mW/cm$^2$<br>• HRT: 20 min<br>• pH: 7 | Initial concentration of contaminant: 2 μM<br>Bisphenol A (BPA): ~80%<br>Carbamazepine (CBZ): ~60% | [99] |

It was also found that generally, UV/$H_2O_2$ processes were less effective in the degradation of CECs as compared to other oxidative species [67,87,88]. Certain CECs are also more resilient to UV/$H_2O_2$ processes. Perfluorooctanoic acid (PFOA), perfluorooctane sulfonate (PFOS), and bisphenol A (BPA) were found to have low degradation performance of <30%, <30%, and <50% respectively. This is likely due to low values of $H_2O_2$ molar absorption coefficient at 254 nm (19.0 M$^{-1}$ cm$^{-1}$) [100], resulting in higher hydrogen peroxide and UV dose for efficient removal.

Many photooxidant processes use persulfate and its derivatives and these were found to be generally greater in performance as compared to UV/$H_2O_2$ processes. For instance, removals of caffeine (60%) and carbamazepine (70%) through UV/$S_2O_3{}^{2-}$ were significantly better than in the UV/$H_2O_2$ [89]. Low quantum yield and photolysis coefficients could be due to the presence of imidazole groups in both caffeine and carbamazepine. However, it was found that persulfate tended to be slightly acidic when used as an oxidant; hence, this might not be suitable for the removal of CECs from treated effluent. Furthermore, radicals generated by persulfate had a lower charge as shown in the Supplementary S3. There are also limited studies on the use of UV/chlorine processes, hence presenting an opportunity for more studies on the use of this aspect, since both UV and chlorine ($Cl_2$) are commonly used as disinfectants in water/wastewater treatment processes, although some studies suggest that disinfection by-products may potentially be formed by these processes [86,87].

Another area that is presently being studied is the use of pre-existing inorganics such as nitrate in wastewater for the promotion of radical species [97,98]. Rizzo [98] found that nitrate as an oxidant could remove CECs with moderate capacity even with the use

of sunlight as an irradiation source. Carbamazepine (CBZ), diclofenac (DCF), and sulfamethoxazole (SMX) were removed from wastewater effluent with 70%, >90%, and 100% efficiency with UVC irradiation, respectively, whereas another study by Zhou et al. [99] showed that nitrates could reduce bisphenol A (BPA) and carbamazepine (CBZ) at an efficiency of ~80% and ~60%, respectively. These results were consistent with the finding from Li et al. [101]. They noted that the presence of $NO_3^-$ promoted the photochemical loss of oxytetracycline (up to 82.9%) in an aqueous solution. The photolysis of $NO_3^-$ leads to the formation of HO• which further promotes the indirect photodegradation of oxytetracycline.

At present, the effects of the inorganics such as nitrates and sulfates in wastewater are still not widely understood and the degradation performance could be studied further for a larger variety of CECs. These oxidants have fewer risks of synergistic impact of the blending of by-products in the effluent stream.

### 2.4. UV/Ozone

The majority of UV/$O_3$ processes as summarized in Table 5 were operated at slightly alkaline pH ($\geq$9), with low-pressure mercury lamps, bench-scale reactors, and an HRT of 20–180 min. UV/$O_3$ showed better removal performance as compared to purely ozonation process with the same treatment condition. Jing et al. [102] reported a better removal of COD and $NH_3$-N in the treatment of atrazine production wastewater (from 2% to 21% respectively in $O_3$ to 55% and 65% respectively in UV/$O_3$). Another study by Xu et al. [103] showed that UV/$O_3$ had better degradation of synthetic wastewater containing sucralose as compared to UV and ozonation alone. Total organic carbon (TOC) removal of 89.8% for UV/$O_3$ as compared to UV and $O_3$ at <5% and 39.1% removal, respectively, was reported. This could be due to a large number of hydroxyl radicals that can be generated in a fast manner which is adequate for the mineralization of CECs. However, it was noted that the combination of ozone and UV process did not improve the degradation of gasoline compounds (benzene, toluene, and isomers of xylene) in comparison with ozone [104].

**Table 5.** Current application of UV/Ozone processes.

| Wastewater/ Compounds | Operation Condition | Removal Efficiency | References |
|---|---|---|---|
| Synthetic wastewater | • UV/$O_3$ (benchtop)<br>• Low-pressure UV lamp (LP-UV)<br>• UV wavelength: 254 nm (UVC)<br>• Power: 125 W<br>• UV intensity: $4.6 \times 10^{-7}$ Einstein/Ls<br>• Ozone dose: 14.7 mg/L<br>• HRT: 52 min<br>• pH: 9.2 | Initial concentration of caffeine: 300 mg/L<br>Color: 99.1% removal<br>Caffeine degradation: 96.5% removal | [105] |
| Synthetic wastewater with sucralose | • UV/$O_3$ (benchtop)<br>• Low-pressure UV lamp (LP-UV)<br>• UV wavelength: 254 nm (UVC)<br>• UV intensity: 33.4 W/m$^2$<br>• Ozone dose: 19.4 mg/L<br>• Ozone flow rate: 35 L/h<br>• HRT: 120 min<br>• pH: 7.0 | Initial concentration of sucralose: 50 mg/L<br>Degradation: 100% (after 30 min)<br>Mineralization: 89.8% (after 2 h) | [103] |

| Wastewater/ Compounds | Operation Condition | Removal Efficiency | References |
|---|---|---|---|
| Real wastewater containing atrazine | • UV/$O_3$ (benchtop)<br>• Low-pressure UV lamp (LP-UV)<br>• UV wavelength: 254 nm (UVC)<br>• Power: $8 \times 3.5$ W<br>• Ozone flow rate: 15 g/h<br>• HRT: 180 min<br>• pH: 12.0 | Initial concentration of atrazine: 0.0232 mM<br>Initial COD: 0.447 M<br>Initial $NH_3 - N$: 1.44 mM<br>Initial $Cl^-$: 5.56 M<br>Atrazine degradation: 95% (after 180 min)<br>COD: 55% removal<br>$NH_3$-N: 65% removal | [102] |
| Groundwater/surface water/secondary effluent spiked with CECs | • UV/$O_3$ (pilot-scale)<br>• Low-pressure UV lamp (LP-UV)<br>• UV wavelength: 254 nm (UVC)<br>• Specific ozone dose: 1.5 mg $O_3$/mg DOC<br>• Ozone flow rate: 0.2 L/min<br>• HRT: 20 min<br>• pH: 8.0–8.2 | Initial concentration of micropollutants: ~150 μg/L each<br>Groundwater: 76–~100% removal<br>Surface water: 84–100% removal<br>Secondary effluent: 89–97% removal | [106] |
| Synthetic wastewater, mixing waste firefighting foam | • UV/$O_3$ (benchtop)<br>• Low-pressure UV lamp (LP-UV)<br>• UV wavelength: 254 nm (UVC)<br>• UV intensity: 83 μW/cm$^2$ (R1), 43 μW/cm$^2$ (R2)<br>• Ozone flow rate: 30 L/min<br>• HRT: 20 min<br>• pH: natural<br>• UV/$O_3$ (pilot-scale)<br>• Low-pressure UV lamp (LP-UV)<br>• UV wavelength: 254 nm (UVC)<br>• Ozone flow rate: 160 L/min<br>• HRT: 20 min<br>• pH: natural | Initial concentration of PFAS: 3–10 μg/L (benchtop)<br>3.15 μg/L (pilot plant)<br>Benchtop: 73% removal<br>Pilot plant: 73% removal | [107] |
| Synthetic wastewater with P-nitroaniline and coal washing plant wastewater | • UV/$O_3$ (benchtop)<br>• Low-pressure UV lamp (LP-UV)<br>• UV wavelength: 254 nm (UVC)<br>• Power: 15 V<br>• UV intensity: 254 mW/m$^3$<br>• Ozone flow rate: 0.9 g/h<br>• HRT: 40 min<br>• pH: 9.0 | Initial concentration of nitroaniline: 10–25 mg/L<br>PNA: 81% removal<br>TOC: 81% removal | [108] |
| Synthetic wastewater | • VUV/$O_3$ (benchtop)<br>• Vacuum-UV lamp (VUV)<br>• UV wavelength: 185 nm (VUV)<br>• Power: 40 W<br>• Ozone flow rate: 1.16 mg/min<br>• HRT: 90 min<br>• pH: 10.0 | Initial concentration:<br>Diethyl dithiocarbonate collector = 100 mg/L<br>Degradation: 99.55% removal<br>TOC: 34% removal | [109] |

Despite the benefit of the enhanced degradation rate, there has been less focus on the study of the UV/$O_3$ process, due to the higher cost of treatment. Ozonation and UV processes are widely known to be energy-intensive and as such the upscaling UV/$O_3$ might not be viable. Furthermore, ozonation requires pH restriction to perform optimally [110], which might not be suitable for CEC treatment. CECs are concentrated mostly in the effluent stream of wastewater treatment facilities which generally have a neutral pH. As

such, pH adjustment needs to be done before and after the UV/$O_3$ process for effluent discharge, increasing treatment costs.

*2.5. Photo-Fenton*

Photo-Fenton processes have been used in a wide variety of CECs degradation of various scales and modes of operations, as seen in Table 6. The majority of the photo Fenton processes uses lower energy UV wavelengths and even solar-powered systems. Degradation time also varies from 10–180 min, depending on the compounds being degraded. Most of the investigations using photo-Fenton processes focus on the removal of target compounds spiked in synthetic matrices at laboratory scale and pilot scales with a good degree of degradation of between 80–90% of CECs.

**Table 6.** Current application of photo-Fenton processes.

| Category | Wastewater/ Compounds | Operation Condition | Removal Efficiency | References |
|---|---|---|---|---|
| Heterogenous | Pharmaceutical wastewater | • UV/Fe<br>• UV wavelength: 190–900 nm (UVC)<br>• Catalyst dose: 1 g/L<br>• Power: 600 W/$m^2$<br>• HRT: 60 min<br>• pH: 3.5 | Initial concentration of contaminant: 5 mg/L<br>ACT removal: ~90%<br>ACE removal: ~90% | [111] |
| | Synthetic water containing ofloxacin (OFL) | • UV/bio-FeMnOx<br>• UV wavelength: <420 nm (sunlight/UVA)<br>• Low power mercury lamp<br>• Chemical dose: stoichiometric amount [$H_2O_2$], 5 mg/L [bio-FeMnOx]<br>• UV power: 500 W<br>• HRT: 90 min | Initial concentration of OFL: 30 mg/L<br>Ofloxacin (OFL) removal: ~90% (UV) | [112] |
| | Synthetic wastewater containing tetracycline (TC) | • UV/nZVI (benchtop)<br>• Low-pressure UV lamp (LP-UV)<br>• UV wavelength: 254 nm (UVC)<br>• Power: 18 W<br>• Lamp intensity = 2500 mcW/$cm^2$<br>• Catalyst dose: 5 mg/L<br>• HRT: 200 min<br>• pH: 9.0 | Initial concentration of TC: 10 mg/L<br>TC: 96.71% removal | [78] |
| | Synthetic wastewater containing amoxicillin (AMX) | • UV/$Fe_3O_4$/g-$C_3N_4$ (benchtop)<br>• Low-pressure UV lamp (LP-UV)<br>• UV wavelength: 254 nm (UVC)<br>• Power: 10 W<br>• Lamp intensity = 3.5 mJ/$cm^2$<br>• Catalyst dose: 1 g/L<br>• HRT: 200 min<br>• pH: Natural | Initial concentration of AMX: 0.25 mMAMX: 89% removal<br>TOC: 60% removal | [61] |

**Table 6.** *Cont.*

| Category | Wastewater/ Compounds | Operation Condition | Removal Efficiency | References |
|---|---|---|---|---|
| | Synthetic wastewater containing oxytetracycline (OTC) and ampicillin (AMP) | • $UV/Fe_3O_4/Bi_2WO_6$ (benchtop)<br>• UV wavelength: Visible<br>• Power: 1000 W<br>• Light intensity = $35 \times 10^3 \pm 1000$ lx<br>• Catalyst dose: 0.5 g/L<br>• HRT: 120 min (OTC), 60 min (AMP)<br>• pH: 6.0 | Initial concentration of OTC and AMP: 0.1 mM each<br>OTC: 73% removal (after 1 h)<br>COD: 62% removal (after 10 h)<br>AMP: 73% removal (after 1 h)<br>COD: 60% removal (after 10 h) | [113] |
| | Synthetic wastewater containing metronidazole (MNZ) | • UV/nZVI (benchtop)<br>• High-pressure UV lamp (HP-UV)<br>• UV wavelength: 365 nm (UVA)<br>• Power: 160 W<br>• Catalyst dose: 0.2 g/L<br>• HRT: 80 min<br>• pH: Natural | Initial concentration of metronidazole: 35 mg/L<br>MNZ: 71.98% removal | [114] |
| | Synthetic wastewater containing tetracycline (TC) | • $UV/(Fe_3O_4/CuO/C)$ (benchtop)<br>• High-pressure UV lamp (HP-UV)<br>• UV wavelength: 420 nm (UVA)<br>• Power: 100 W<br>• Catalyst dose: 3 g/L<br>• HRT: 60 min<br>• pH: 7 | Initial concentration of TC: 50 mg/L<br>TC: ~90% removal | [115] |
| Homogenous | Synthetic wastewater containing an anti-inflammatory substance | • UV/Fe<br>• UV wavelength: 289 and 367 nm (UVC)<br>• Chemical dose: 400 mg/L of $[H_2O_2]$, 1.75 mg/L of [Fe]<br>• HRT: 120 min<br>• pH: 3–4 | Initial concentration of contaminant: 10 mg/L (Ketoprofen, tenoxicam, and meloxicam drugs)<br>COD removal: 98% removal | [116] |
| | Real wastewater | • UV/Fe-EDDS<br>• UV wavelength: 327–384 nm (UVC)<br>• Chemical dose: 0.88, 0.1 mM $Fe^{3+}$<br>• Power: 30 W m$^{-2}$<br>• HRT: 15 min<br>• pH: 7–8 | Initial concentration of contaminant: 5000–50,000 ng/L (total CECs)<br>COD removal: <80% removal | [117] |
| | Real wastewater | • UV/Fe-EDDS<br>• UV wavelength: 327–384 nm (sunlight)<br>• Chemical dose: 0.88 mM $[H_2O_2]$ 0.1 mM $[Fe^{3+}$-EDDS]<br>• HRT: 30 min<br>• pH: 7–8 | Initial concentration of contaminant: na<br>Total EC removal: >80% | [118] |

**Table 6.** *Cont.*

| Category | Wastewater/ Compounds | Operation Condition | Removal Efficiency | References |
|---|---|---|---|---|
| | Pharmaceuticals wastewater | • UV/Fe-oxalic + sonication<br>• UV wavelength: 254 nm<br>• Chemical dose: 5 mg/L [Goethite $\alpha$-FeOOH], 10 mmol/L [$H_2O_2$]<br>• HRT: 5–60 min<br>• pH: 7.84 | Initial concentration of contaminant: 12 mg/L (SDZ), 5.8 mg/L (TOC) sulfadiazine (SDZ) removal: ~100%(5 min), ~80%(60 min) | [119] |
| | Real wastewater | • Solar/Fe-EDDS<br>• UV wavelength: 327 to 384 nm (sunlight)<br>• Chemical dose: 0.88 [$H_2O_2$], 0.1 mM [Fe]<br>• HRT: 40 min<br>• pH: neutral | Initial concentration of CECs: 59.1–77.7 ng/L Total removal: ~61% (20 min) | [120] |
| | Real wastewater | • Solar/Fe- EDDS<br>• UV wavelength: 327 to 384 nm (sunlight)<br>• Chemical dose: 3 ppm [$Fe_2(SO_4)_3$] and 2.75 ppm [$H_2O_2$]<br>• HRT: 60 min<br>• pH: 7–8 | Initial concentration of contaminant: 1 ppm Total removal (DI): 89–94% (AMX), 92–95% (PC) Total removal (WW): ~50% (AMX), ~30% (PC) | [70] |
| | Real wastewater | • Solar/Fe-EDDS<br>• UV wavelength: 300–800 nm (sunlight/artificial sunlight)<br>• Chemical dose: 0.054 [Fe], 1.47 mM [$H_2O_2$]<br>• HRT: 10–30 min<br>• pH: 7 | Initial concentration of contaminant: 100 µg/L Caffeine removal: ~90% Carbamazepine removal: ~90% Diclofenac removal: ~90% Sulfamethoxazole removal: ~90% Trimethoprim removal: ~90% | [121] |
| | Real wastewater doped with SMX | • UV/Fe-EDDS, solar/Fe-EDDS,<br>• UV wavelength: 327–384 nm (sunlight/UVA)<br>• Chemical dose: 0. [$Fe^{3+}$-EDDS], 0.88 mM [$H_2O_2$]<br>• UV Power: 30 W/m$^2$<br>• HRT: 60 min<br>• pH: 6.5–7.5 | Initial concentration of contaminant: 50 µg/L Sulfamethoxazole removal: ~80% (UV), ~40% (solar) Wild *E. coli* inactivation: below the detection limit | [122] |
| | Real wastewater doped with SMX and IMD | • Solar/Fe-NTA<br>• UV wavelength: 327–384 nm (sunlight/UVA)<br>• Chemical dose: 0.1 [$Fe^{3+}$-EDDS], 0.88 mM [$H_2O_2$]<br>• UV Power: 35 ± 2 W/m$^2$<br>• HRT: 60 min<br>• pH: 6–7 | Initial concentration of contaminant: 50 µg/L Sulfamethoxazole removal: ~90% (UV) Imidacloprid removal: ~80% (UV) | [123] |

**Table 6.** *Cont.*

| Category | Wastewater/ Compounds | Operation Condition | Removal Efficiency | References |
|---|---|---|---|---|
| | PPCPs removal in wastewater effluent | • UV/FeIII-NTA<br>• UV wavelength: 365 nm (sunlight/UVA)<br>• Low-power mercury lamp<br>• Chemical dose: 4.54 [$H_2O_2$], 0.178 mM [FeIII-NTA]<br>• UV Power: 4.05 mW/cm$^2$<br>• HRT: 60 min<br>• pH: 7 | Initial concentration of contaminant: 452.6 (CBZ), 394.6 (CRMT), and 101.1 (IBP) μg/L CBZ removal: ~80% Crotamiton (CRMT): ~80% Ibuprofen (IBP): ~80% | [124] |
| | PPCPs removal in wastewater effluent | • UV/Fe-HA<br>• UV wavelength: 295–400 nm (artificial sunlight)<br>• Low-power mercury lamp<br>• Chemical dose: 1.0 [$H_2O_2$], 100 mg/L [FeIII-NTA]<br>• UV Power: 1500 W, cut-off filter at below 340 nm<br>• HRT: 60 min<br>• pH: 3 | Initial concentration of contaminant: 0.2, 0.1 mM carbamazepine, 20 μmol (blended) Ibuprofen removal: ~30% Bisphenol A removal: ~80% tolylbenzotriazole removal: ~90% Carbamazepine removal: ~80% Blended mix: ~90% (in DI water) | [125] |
| | PPCPs removal in wastewater effluent | • UV/Fe<br>• UV wavelength: natural sunlight<br>• Chemical dose: 50 mg/L [$H_2O_2$], 20 mg/L [Fe]<br>• UV Power: 21–26.5 W/m$^2$<br>• HRT: 180 min<br>• pH: neutral | Initial concentration of contaminant: as per wastewater stream Blended: 73–82% Azithromycin (AZT) removal: 24% Ciprofloxacin (CIP) removal: 100% Clarithromycin (CLR) removal: 8–24% Clindamycin: 57–86% Enrofloxacin (ERF) removal: 100% Erythromycin (ENR) removal: 22–36% levofloxacin (LEV) removal: 61–75% Lincomycin (LIN) removal: 84% Metronidazole (MET) removal: 70% | [126] |

Notable publications describing the effects of chemical structures on the efficiency of photo/Fenton-based systems are summarized above. Recent innovation with magnetic carbon-based heterogenous composites addresses the reusability and separation of conventional heterogenous catalysts. Alani et al. [115] reported that the photocatalytic performance of the magnetic catalyst remains relatively high even after 5 consecutive uses. Wu et al. [127] investigated the photo-Fenton degradation for methylene blue, methyl orange, rhodamine B, 2,4-chlorophenol, and bisphenol A with carbon quantum dots on the iron-based catalyst α-FeOOH. Methyl orange showed the lowest degradation (~50%) after 20 min of reaction. This could be due to the electrostatic repulsive force between methyl orange and the catalyst which limiting the adsorption. BPA and 2,4-chlorophenol showed good degradation (~100%) due to the π–π interaction between these compounds and car-

bon quantum dots. Different CECs may have different removal mechanisms during the photo-Fenton process. For instance, Guerra et al. [70] proposed degradation pathways for paracetamol and amoxicillin in the photo-Fenton solar process. With different functional groups such as amino, carbonyl, and hydroxyl groups, paracetamol may undergo oxidative attacks: aromatic hydroxylation, allylic and benzylic oxidations, amine dealkylation, and amine oxidation. Amoxicillin has more functional groups in its chemical structure compared to paracetamol. The degradation of amoxicillin would form amoxilloic acid resulting from the hydroxylation of the aromatic ring and the opening of the four-membered β-lactam ring. Conventional Fenton processes require strict pH control at acidic ranges which might not be a suitable process for downstream wastewater effluent, since the pH of the effluent is usually neutral [117,118,120,121]. Inhibition of the Fenton process due to the solubility of the iron species can be circumvented with the use of ligands where a large number of recent studies focus on the use of ligands like EDDS to operate the reactors at a neutral pH. Generally, the operation pH is at pH 7, with some studies running it at pH 3 [111,116]. More research is on homogenous Fenton, which has a higher reaction kinetics as compared to heterogeneous Fenton. Heterogenous photo Fenton also has issues with the retention of catalysts. Wang et al. reported that the nZVI catalyst lost 15% of its initial amount after 5 consecutive runs [114], due to the dissolution of iron into the solution.

The recent innovation of the use of solar-assisted Fenton, raceway pond reactors, shows promising results for the degradation of CECs [113,117,119,121,122]. This reactor configuration requires less energy input due to the use of solar energy and also shows a high degree of degradation. The height of the race pond bed seems to contribute significantly to the degradation performance of CECs. More recently, alternative and cheaper ligands such as NTA are also being test bedded against conventional EDDS ligands [123]. At present, the study of this reactor is still in its infancy stage and hence has the potential to be developed further.

### 2.6. Photocatalysis

Photocatalysis uses solar energy/UV lamps with a catalyst and has been used in a wide variety of CECs degradation of various scales and modes of operations. The majority of the photocatalysis processes utilize both lower energy UV wavelengths and even solar energy. Degradation time also varies from 10–180 min, depending on the compounds being degraded. Most of the investigations using photocatalysis processes have been focused on the removal of target compounds spiked in synthetic matrices at a laboratory scale with a good degree of degradation of between 80–90% of CECs. Various types of photocatalysts such as titanium dioxide ($TiO_2$), zinc oxide ($ZnO$), tungsten trioxide ($WO_3$), and graphitic carbon nitrides ($g$-$C_3N_4$) have been explored and are summarized in Table 7.

**Table 7.** Current application of photocatalysis processes.

| Categories | Wastewater/ Compounds | Operation Condition | Removal Efficiency | References |
|---|---|---|---|---|
| Titanium oxide-based catalyst | Synthetic wastewater containing tris-(2-chloroisopropyl) phosphate (TCPP) | • $UV/TiO_2$ (benchtop) <br> • Low-pressure UV lamp (LP-UV) <br> • UV wavelength: 254 nm (UVC) <br> • Power: 15 W <br> • UV fluence: 4.7 mW/cm$^2$ <br> • Catalyst dose: 0.1 g/L <br> • HRT: 60 min <br> • pH: 7.0 | Initial concentration of contaminant: 1 mg/L TCPP: ~100% removal TOC: ~80% removal | [128] |

**Table 7.** *Cont.*

| Categories | Wastewater/ Compounds | Operation Condition | Removal Efficiency | References |
|---|---|---|---|---|
| | Synthetic wastewater with acesulfame potassium (ACE-K) and sodium saccharin (SAC) | <ul><li>UV/$TiO_2$ (benchtop)</li><li>High-pressure UV lamp (HP-UV)</li><li>UV wavelength: 365 nm (UVA)</li><li>Power: 125 W</li><li>Photon flux rate: $3.435 \times 10^{-7}$ Einstein/Ls</li><li>Catalyst dose: 0.375 g/L</li><li>HRT: 60 min</li><li>pH: 6.0</li></ul> | Initial concentration of contaminant: 20 mg/L each Degradation: Both: 100% removal Mineralization: ACE-K: 57% removal SAC: 49% removal | [129] |
| | Synthetic wastewater containing tetracycline hydrochloride (TC) | <ul><li>UV/$TiO_2$ (benchtop)</li><li>High-pressure UV lamp (HP-UV)</li><li>UV wavelength: 360 nm (UVA)</li><li>Power: 30 W</li><li>Catalyst dose: 0.12 g</li><li>HRT: 240 min</li><li>• pH: 4.0</li></ul> | Initial concentration of contaminant: 40 mg/L TC: 96% removal | [130] |
| | Synthetic wastewater containing sulfamethazine (SMT) | <ul><li>UV/$TiO_2$(P25) (benchtop)</li><li>High-pressure UV lamp (HP-UV)</li><li>UV wavelength: 360 nm (UVA)</li><li>Power: 9 W</li><li>Catalyst dose: 0.25 g/L</li><li>HRT: 240 min</li><li>pH: 6.0</li></ul> | Initial concentration of contaminant: 50 mg/L SMT: 42% removal COD: 34% removal | [131] |
| | Secondary treatment effluent | <ul><li>UV/$TiO_2$(P25) (Benchtop)</li><li>High-pressure UV lamp (HP-UV)</li><li>UV wavelength: 417 nm (UVA)</li><li>UV intensity: 400–500 W/$m^2$</li><li>Catalyst dose: 1.0 g/L</li><li>HRT: 240 min</li><li>pH: natural</li></ul> | Initial concentration of contaminants: varied Contaminants: 25–90% removal | [132] |
| | Wastewater effluent or synthetic wastewater containing Bisphenol A | <ul><li>UV/$TiO_2$ and variants (Benchtop)</li><li>Solar simulator</li><li>UV wavelength: Visible</li><li>Power: 500 W</li><li>Catalyst dose: 0.5 g/L</li><li>HRT: 120 min</li><li>pH: natural</li></ul> | Initial concentration of BPA: 10 mg/L Synthetic wastewater: With UV/$TiO_2$ (P25): 100% removal With UV/$TiO_2$-$WO_3$: 30% removal With UV/$TiO_2$: 38% removal Real wastewater: With UV/$TiO_2$ (P25): 62% removal With UV/$TiO_2$-$WO_3$: 2% removal | [57] |
| | Synthetic wastewater with dinitro butyl-phenol (DNBP) (herbicide) | <ul><li>UV/$TiO_2$</li><li>Xe lamp</li><li>UV wavelength: 420 nm (UVA)</li><li>Power: 500 W</li><li>Catalyst dose: 0.5 g/L</li><li>HRT: 240 min</li><li>pH: natural</li></ul> | Initial concentration of DNBP: 20 mg/L DNBP: 28% removal | [133] |

**Table 7.** *Cont.*

| Categories | Wastewater/ Compounds | Operation Condition | Removal Efficiency | References |
|---|---|---|---|---|
| | Synthetic wastewater containing carbamazepine (CBZ), ibuprofen (IBU,) and sulfamethoxazole (SMX) | • UV/TiO$_2$(P25) (benchtop) <br> • High-pressure UV lamp (HP-UV) <br> • UV wavelength: 365 nm (UVA) <br> • Power: 160 W <br> • Catalyst dose: Bundle of 30 fibers <br> • HRT: 180 min <br> • pH: natural | Initial concentration of pharmaceuticals: 5 mg/L each <br> IBU: 38% removal <br> CBZ: 38% removal <br> SMX: 64% removal | [134] |
| | Synthetic wastewater containing metronidazole (MNZ) | • UV/TiO$_2$(P25) (benchtop) <br> • Low-pressure UV lamp (LP-UV) <br> • UV wavelength: 254 nm (UVC) <br> • Power: 20 W <br> • Catalyst dose: 0.2 g/L <br> • HRT: 80 min <br> • pH: natural | Initial concentration of MNZ: 35 mg/L <br> MNZ: 43.02% removal | [114] |
| | Synthetic wastewater containing saccharin (SAC) | • UV/TiO$_2$(P25) (benchtop) <br> • LED lamp <br> • UV wavelength: 365 nm (UVA) <br> • Power: 11 W <br> • Catalyst dose: 0.125 g/L <br> • HRT: 45 min <br> • pH: 4.6 | Initial concentration of saccharin: 5 mg/L <br> SAC: 100% removal (after 30 min) | [135] |
| Zinc based catalyst | Synthetic wastewater containing sulfamethazine (SMT) | • UV/ZnO (benchtop) <br> • High-pressure UV lamp (HP-UV) <br> • UV wavelength: 360 nm (UVA) <br> • Power: 9 W <br> • Catalyst dose: 0.25 g/L <br> • HRT: 240 min <br> • pH: 6.0 | Initial concentration of contaminant: 50 mg/L <br> SMT: 64% removal <br> COD: 45% removal | [131] |
| | Synthetic wastewater | • UV/ZnO (benchtop) <br> • High-pressure UV lamp (HP-UV) <br> • UV wavelength: 366 nm (UVA) <br> • Power: 2 × 8 W <br> • Catalyst dose: 0.2 g/L <br> • HRT: 120 min <br> • pH: 7.0 <br> • UV/ZnO (pilot plant) <br> • Natural sunlight <br> • UV wavelength: VIS + NIR, UVA, UVB, and UVC <br> • UV Intensity: 0.2 ± 0.1–1011.6 ± 66.2 iW/m$^2$ <br> • Catalyst dose: 0.2 g/L <br> • HRT: 240 min <br> • pH: 7.2 | Initial concentration of contaminants: 0.3 mg/L <br> Synthetic wastewater: Contaminants: 97.5–99.7% removal (after 90 min) <br> Real wastewater: Contaminants: 76.0–100.0% removal (after 240 min) | [59] |
| | Synthetic wastewater with sulfathiazole (STZ) | • UV/ZnO (benchtop) <br> • Xe lamp <br> • UV wavelength: 350–400 nm (UVA) <br> • Power: 1000 W <br> • Catalyst dose: 2 g/L <br> • HRT: 90 min <br> • pH: natural | Initial concentration of STZ: 0.1 M <br> STZ: 69% removal | [66] |

**Table 7.** *Cont.*

| Categories | Wastewater/ Compounds | Operation Condition | Removal Efficiency | References |
|---|---|---|---|---|
| | Synthetic wastewater containing methylene blue (MB) and real industrial dye | • UV/ZnO (benchtop)<br>• Sunlight<br>• UV wavelength: visible<br>• Catalyst dose: 0.2 g/L<br>• HRT: 80 min<br>• pH: natural | Initial concentration of MB: 20 mg/L MB: ~99% removal (after 120 min) | [136] |
| | Synthetic wastewater containing phenol and dinitrophenol | • UV/SV/FG24 (benchtop)<br>• LED lamp<br>• UV wavelength: visible light<br>• Power: 35 W<br>• UV intensity: 750 lx<br>• Catalyst dose: 0.5 g/L<br>• HRT: 180 min<br>• pH: 4.0 | Initial concentration of phenol and DNP: 0.1 mM each<br>Phenol: 95% removal<br>DNP: 88% removal | [137] |
| | Synthetic wastewater containing amoxicillin (AMX) | • UV/g-$C_3N_4$ (benchtop)<br>• Low-pressure UV lamp (LP-UV)<br>• UV wavelength: 254 nm (UVC)<br>• Power: 10 W<br>• Lamp intensity = 3.5 mJ/$cm^2$<br>• Catalyst dose: 1 g/L<br>• HRT: 200 min<br>• pH: natural | Initial concentration of AMX: 0.25 mM<br>AMX: 67% removal<br>TOC: 42% removal | [61] |
| Carbon-based catalyst | Synthetic wastewater with diclofenac sodium (DCF) | • UV/0.20%$Co_3O_4$-g-$C_3N_4$ (benchtop)<br>• Xe lamp<br>• UV wavelength: 420 nm (visible)<br>• Power: 50 W<br>• Catalyst dose: 0.5 g/L<br>• HRT: 120 min<br>• pH: natural | Initial concentration of DCF: 10 mg/L<br>DCF: 100% removal (30 min in presence of 0.1 mM PMS) | [138] |
| | Synthetic wastewater containing carbamazepine, ibuprofen, and sulfamethoxazole | • UV/$TiO_2$_2.7% rGO (benchtop)<br>• High-pressure UV lamp (HP-UV)<br>• UV wavelength: 365 nm (UVA)<br>• Power: 160 W<br>• Catalyst dose: bundle of 30 fibers<br>• HRT: 180 min<br>• pH: natural | Initial concentration of pharmaceuticals: 5 mg/L each<br>IBU: 81% removal<br>IBU-TOC: 52% removal<br>CBZ: 54% removal<br>CBZ-TOC = 54% removal<br>SMX: 92% removal<br>SMX-TOC = 59% removal | [134] |
| | Synthetic wastewater containing phenol and dinitrophenol (DNP) | • UV/FG24 (benchtop)<br>• LED lamp<br>• UV wavelength: visible light<br>• Power: 35 W<br>• UV intensity: 750 lx<br>• Catalyst dose: 0.5 g/L<br>• HRT: 180 min<br>• pH: 4.0 | Initial concentration of phenol and DNP: 0.1 mM each<br>Phenol: 36% removal<br>DNP: 25% removal | [137] |
| | Synthetic wastewater containing 4-nitrophenol (PNP) | • UV/3%rGO/$ZrO_2$/$Ag_3PO_4$ (benchtop)<br>• Low-pressure UV lamp (LP-UV)<br>• UV wavelength: 254 nm (UVC)<br>• Power: 4 × 6 W<br>• Catalyst dose: 0.25 g/L<br>• HRT: 120 min<br>• pH: 6.0 | Initial concentration of PNP: 15 mg/L<br>PNP: 97% removal | [139] |



**Table 7.** *Cont.*

| Categories | Wastewater/ Compounds | Operation Condition | Removal Efficiency | References |
|---|---|---|---|---|
| Other types of catalyst | Synthetic wastewater and treated sewage effluent spiked with ibuprofen (IBU) | • UV/gCTFS (benchtop) <br> • Fluorescent lamp <br> • UV wavelength: visible <br> • Power: $8 \times 8$ W <br> • Lamp intensity = 330 W/m$^2$ <br> • Catalyst dose: 1 g/L (syn wastewater), 2 g/L (real wastewater) <br> • HRT: 180 min <br> • pH: natural (syn wastewater), $6.9 \pm 0.2$ (real wastewater) | Initial concentration of IBU: 2 mg/L Synthetic wastewater: IBU: 97% removal (15 min) Real wastewater: IBU: 92% removal (180 min) | [140] |
| | Synthetic wastewater with sulfathiazole (STZ) | • UV/LuAG: Ce/ZnO (benchtop) <br> • Xe lamp <br> • UV wavelength: 350–400 nm (UVA) <br> • Power: 1000 W <br> • Catalyst dose: 2 g/L <br> • HRT: 90 min <br> • pH: natural | Initial concentration of STZ: 0.1 M STZ: ~100% removal | [66] |
| | Synthetic wastewater with 1,4-dioxane (1,4-D) | • Solar/WO$_3$/n$\gamma$-Al$_2$O$_3$ (benchtop) <br> • Solar simulator <br> • Wavelength: 190–1100 nm <br> • Power: 40 mW <br> • Catalyst dose: 100–700 mg/L <br> • HRT: 4 h <br> • pH: 6.8 | Initial concentration of 1,4-D: 50 mg/L 1,4-D: >75% mineralization | [141] |
| | Synthetic wastewater with levofloxacin (LFX) and ketoprofen (KPT) | • Photoelectrochemical/$\beta$25 modified WO$_3$ (benchtop) <br> • Hg medium pressure lamp <br> • Wavelength: 360–380 nm <br> • Power: 0.128 W <br> • Catalyst dose: 0.1 g <br> • HRT: 5 h <br> • pH: 6.8 | Initial concentration of LFX: 10 mg/L LFX: ~90% removal Initial concentration of KPT: 10 mg/L KPT: >65% removal | [142] |
| | Synthetic wastewater with dexamethasone (DXM) | • BLB/WO$_3$ (benchtop) <br> • UV and halogen lamps <br> • Wavelength: 254 nm (UVC), 365 nm (UVA) and >380 nm (halogen) <br> • Power: 0.128 W <br> • Catalyst dose: 500 mg/L <br> • HRT: 60 min <br> • pH: 3 | Initial concentration of DXM: 5 mg/L DXM: ~100% removal | [143] |

The degradation rate of CECs by photocatalysis is found to be closely related to their molecular structures. Eskandarian [68] found that decomposition kinetics of CECs by TiO$_2$ photocatalytic followed the order: sulfamethoxazole > diclofenac > ibuprofen > acetaminophen. Sulfamethoxazole is highly reactive due to the NH group in its chemical structure. Ibuprofen could be decomposed via rearrangement of the acidic group, followed by decarboxylation reaction and dehydrogenation. However, it is less flexible in degradation sites due to its molecular structure. Degradation of acetaminophen is the most difficult of the four compounds. The mechanism involved the removal of the amide group (CH$_3$CONH), formation of phenoxy radical that will react with superoxide radical. In another study, Alverez–Corena et al. [144] found the decreasing trend of UV/TiO$_2$ degradation kinetics for 5 CECs: Gemfibrozil > 17$\beta$ estradiol > N-nitrosodimethylamine (NDMA) > 1,4-dioxane > tris-2-chloroethyl phosphate (TCEP). The high degradation for gemfibrozil

could be attributed to the presence of a deprotonated carboxyl group in its structure which can enhance its adsorption capacity on the photocatalyst surface. N-NO bond in NDMA could act as an electron donor to the $TiO_2$ surfaces. C-O bonds in 1,4-dioxane could be served as hydrogen bond acceptors for dipolar attractions. TCEP is without ionizable functional groups in its structure. In addition, a high pKa of 14.86 of its leaving group 2-chloroethanol and higher dipole moment makes it difficult to be removed. Hence, TCEP showed the slowest degradation rate.

Conventionally, UVC wavelengths are used in photocatalysis especially for $TiO_2$ based catalysts [140]. However, in recent years there has been a greater push for other lower energy-intensive wavelengths, such as visible light and solar power for the degradation of CECs. Morphological modification of $TiO_2$ based catalyst has allowed for visible light to be used as an irradiation source [57]. An alternative catalyst such as zinc-based catalyst is also explored that is also able to use sunlight as an irradiation source [59,116]. However, the degradation performance is not as comparable to $TiO_2$ based catalysts. A potential attractive perspective is the use of bimetallic catalyst which combines the efficiency of $TiO_2$ based catalyst and the solar capabilities of zinc-based catalyst.

In most applications of photocatalysts, nano-catalysts are used due to their high surface-to-volume ratio. However, the recovery of this nano-catalyst after usage has been an ongoing issue bottlenecking the large-scale application of these processes [61,135,139]. To overcome this problem, recent researchers have incorporated magnetic elements such as magnetite so that resultant photocatalyst can be easily retrieved from the treated effluent [57,61], but this comes at the expense of adsorptive capacity as reported by Juang et al. [145], wherein the adsorption capacity of AC is reduced by 15% after doping $Fe_3O_4$ to achieve the magnetic capability. Another issue with the formed catalyst is the reusability of the catalyst. Kumar et al. [140] reported that modified $TiO_2$ has a good separation of formed catalysts and high removal of ibuprofen even when reused 3 times. This is similarly reported for the heterogenous photo-Fenton process. However, the current study on the long-term usage of such catalysts is still in its infancy stages and presents the potential for future development. The coupling of solar and magnetite into conventional photocatalytic processes has scaled-up potential and presents an opportunity for future development.

Newer materials such as $WO_3$ and carbon-based photocatalyst have also shown promising results in the degradation of CECs. However, these catalysts were mostly demonstrated for the use of degradation of a singular compound in synthetic wastewater. The cost consideration for the use of these catalysts was also not well evaluated in the literature, due to its novelty. Further research can be done to evaluate its suitability for CECs degradation in real wastewater.

### 2.7. Light-Driven Detection of CEC in Treatment Systems

A cheaper alternative for the detection of CECs is through the use of light-based spectroscopy. Light-driven detection techniques have been widely proven to be useful in characterizing natural organic matter (NOM) in natural water, drinking water, and wastewater. Light-driven detection methods could be broadly divided into 3 segments that have different mechanisms (Table 8). Adsorption of light measures the 'missing' wavelength of light that is shone on the water sample [146]. Light-driven detection is based on the absorption of light by organic compounds which results in the excitation of the electrons from the ground state to a higher energy state. The energy difference of each ground state and excitation state pair corresponds to an absorption band. Compounds that contain aromatic rings and double bonds can absorb energy in the form of ultraviolet light to excite the electrons to higher anti-bonding molecular orbitals. UV absorbance, especially at 254 nm, is one of the most widely used surrogate parameters to quantify NOM reactivity. Chon et al. demonstrated a piecewise linear correlation between the differential of $UV_{254}$ and the elimination of 17$\alpha$-ethynylestradiol (EE2), carbamazepine, atenolol, bezafibrate, ibuprofen, and p-chlorobenzoic acid [147]. On the other hand, Pisarenko et al.

observed significant correlations between reduction in $UV_{254}$ and removal of CEC by ozonation including tris-2-chloroethyl phosphate, meprobamate, primidone, meta-N, N-diethyl toluamide, phenytoin, trimethoprim, sulfamethoxazole, and carbamazepine with $R^2$ ranged from 0.778 to 1.000 [148]. These CECs are electron-rich compounds, e.g., contain aromatic rings or unsaturated carbon bonds (double or triple) in their molecular structure. Previous studies also found that •OH radicals tend to react with large molecules with various reaction sites, aromatic compounds, electron-rich organic moieties [149]. Hence, UV254 or UV280 nm could be used as indicators to predict the removal of CECs from $UV/H_2O_2$. Newer forms of chemiluminescence use enzyme-linked immunosorbent assays (ELISA) and time-resolved fluoroimmunoassays to detect CECs [150] with a high degree of accuracy, whereas for the excitation and emission type, a known light source is shone on the sample and the measured signals would be the excitation wavelength vs. emission wavelength vs. fluorescence intensity [151]. The EEM could be used to discriminate different groups of NOM based on the difference in light emission and excitation of fluorophores. NOMs with certain molecular structures are reported to have fluorescent properties in a wide range of excitation/emission wavelengths [152]. Hence, samples with different NOM have distinct features with maxima located at characteristic combinations of excitation and emission wavelengths. These EEM features could be used to predict the removal of CECs. Lastly, the physicochemical signal detection relies on the physio-chemical reaction to light onto the sample and is highly dependent on the compounds being monitored [153]. Infrared spectroscopy was also explored to detect a low concentration of CECs. Quintelas et al. demonstrated the use of Fourier transform near-infrared (FT-NIR) spectroscopy for the detection of ibuprofen, sulfamethoxazole, 17β-estradiol, and carbamazepine, with a high degree of accuracy with $R^2 \sim 0.95$ [154], although it was noted that other compounds tested were not as accurate. The analytical signal of this sensor is the refractive index change which was recorded along with concentration change of PFOA from 0 to 4 ppb. Recently, various core-shell nanostructures have been widely developed to enhance the surface-enhanced Raman scattering (SERS) technique for the detection of pesticides [155]. Fang et al. [156] developed an aptamer-based conformation cooperated enzyme-assisted SERS technology to detect CEC. LOD of 4.8 pg/L n aqueous solution was achieved for chloramphenicol. Newer methodology such as surface plasmon resonance (SPR) was also demonstrated to have the potential for CEC detection. Cennamo et al. demonstrated the potential for SPC detection of perfluorinated alkylated substances (PFAs) with an impressive detection limit of 0.13 ppb [157,158].

**Table 8.** Evaluation of light-driven technology for the detection and management of CECs.

| Type of Process | Name of Process | Theory | Mechanism | Advantages | Disadvantages | References |
|---|---|---|---|---|---|---|
| Adsorption of light | FT-NIR spectroscopy | Polychromatic light beam at a sample, measure the intensity of the light as a function of time. It allows simultaneous measurement over the whole wavelength range | Electromagnetic radiation (EMR) interacts with atoms and molecules in discrete ways to produce characteristic absorption profiles $f = \frac{c}{\lambda}$ $E = \frac{hc}{\lambda} \times 10^9$ Planck's constant, h = $6.63 \times 10^{-34}$ Js Speed of light in a vacuum, c = $2.998 \times 10^8$ ms$^{-1}$ | Able to do online analysis | Might not be the most accurate due to similar adsorption spectra of various chemical bonds | [153] |
| | Infrared (IR) spectrum | Beam containing a different combination of frequencies shone on the sample and measured for light absorbed. This process is rapidly repeated with different combinations of light wavelengths. Correlations between the data points would be used to infer contaminants in the sample | | Slightly more accurate due to the use of multiple combined wavelengths | - | [159] |

**Table 8.** *Cont.*

| Type of Process | Name of Process | Theory | Mechanism | Advantages | Disadvantages | References |
|---|---|---|---|---|---|---|
| Excitation and emission | Excitation emission matrix (EEM) | An EEM is a 3D scan, of the excitation wavelength vs. emission wavelength vs. fluorescence intensity of a sample when a given monochromatic beam of light is shone on the sample | Similar to the mechanism above, EEM also uses a similar mechanism, however, it correlates the excitation, reemission and fluoresces as well | Fast and quick analysis | At present might not be as accurate. It requires a large scale of intensive data processing | [151] |
| | Size exclusion coupled UV based detection | Size exclusion coupled detection methods | This method is similar to the EEM analysis except it has an element of size exclusion | More accurate measurements, due to an additional size exclusion mechanism | Might not be as accurate presently | [160] |
| | Immunoassay | Measure changes in color or emission of light | Antibodies competitively capture dissolved targets and immobilized antigens. After the washing step, the labeled secondary antibodies bond to the corresponding antibodies. Signals can be obtained after incubation and another washing procedure | Low cost, simple procedure | Organic solvent and environmental factors may interfere with the immunoreaction | [161] |
| Physio-chemical signal detection | Chemiluminescence | Chemical reactions occur, exhibiting light signatures | Explores the chemical signature of selected types of compounds which gives off fluorescence | Able to identify the luminescence compounds with a fairly high degree of accuracy | Highly selective to types of compounds that emit fluorescence | [162] |
| | Surface-enhanced Raman scattering (SERS) | Novel spectrum analysis technology based on the combination of Raman spectroscopy and nanotechnology | The molecular fingerprint specificity of Raman spectroscopy is used for the detection of CECs. The enhancement Raman signal is achieved through electromagnetic enhancement mechanism and chemical enhancement mechanism | High sensitivity, non-destructive, capability for molecular fingerprint | Not stable, not cost-effective, Complex synthetic procedure, complicated synthetic procedure | [155] |
| | Surface plasmon resonance band | Collective electron charge oscillations of contaminants excited by light are measured | - | Only available for detection for nano-metal detection and costly | Only available for detection for nano-metal detection | [163] |

At present, there is limited usage of these methodologies due to their inaccuracy and selectivity of methods used. The main challenge for this technology depends on the correlation between the CECs and their physical and chemical properties. This correlation is site- and treatment-specific and needs to be calibrated and verified, while considering inner-filtering effects. Future development coupled with machine learning might potentially produce higher accuracy for light-driven detection strategies. Artificial neural network (ANN) modeling, which has been used to model and calibrates complex systems [164,165], shows potential to be coupled with UV-detection strategies. More studies could be done to use such calibration means to enhance current detection methods. To address the selectivity of detection means, categorizing of CECs according to their physical–chemical structures

would give precedence to detection methods for newer CECs and also help with the screening of type of detection methods used.

## 3. Cost–Benefit Analysis of Light-Driven AOPs on the Treatment of CECs

Simulated treatment costs of various light-driven processes are studied and evaluated using a matrix $E_{EO}$ value developed by Bolton et al. [166] (Figure 3). Optimized conditions from the articles reviewed would be set as the operation condition for the computation of the $E_{EO}$ values. Detailed computation of the $E_{EO}$ values can be found in the Supplementary S4.

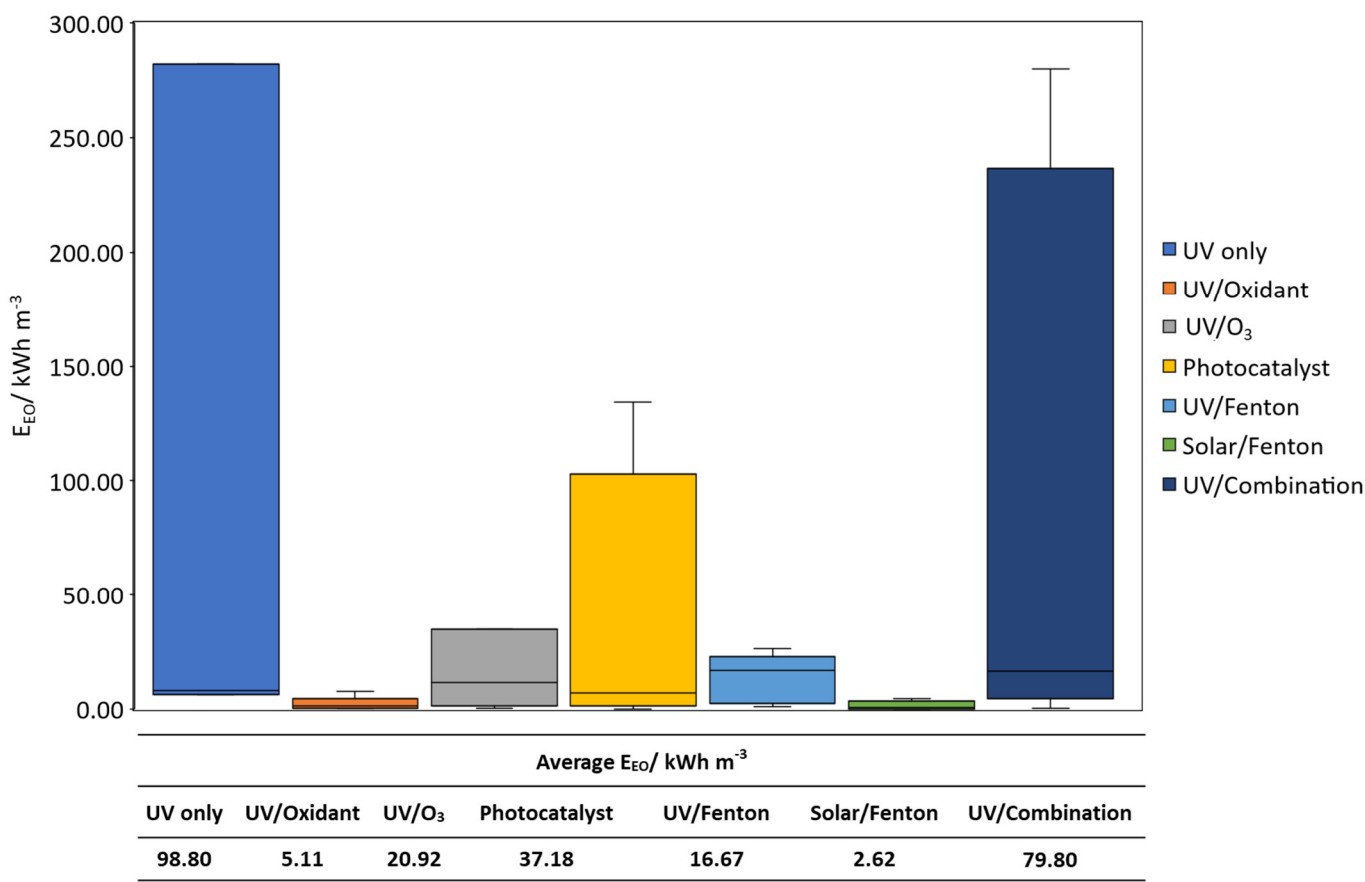

**Figure 3.** Average $E_{EO}$ values of reviewed light-driven AOPs (note: A range of $E_{EO}$ values were shown in the figure as error bars, while the bar graphs showed the average $E_{EO}$ values).

It can be seen that the average $E_{EO}$ values are in this sequence: UV only > UV/combination > photocatalyst > UV/O$_3$ > UV/Fenton > solar/Fenton. The average $E_{EO}$ of the systems computed is 98.8, 5.11, 20.92, 37.18, 16.67, 2.62, and 76.80 kWh m$^{-3}$ for UV only, UV/oxidant, UV/O$_3$, photocatalyst, UV/Fenton, solar/Fenton, and UV/combination, respectively. UV and UV/oxidant strength and selectivity of oxidants result in certain CECs having better degradation than others [153], hence resulting in a varied computation. It was also noted that for selected studies relatively low initial concentration (<0.01 mg/L) of CECs used in the study for UV/oxidant, compared to other processes also likely contributed to the low $E_{EO}$ values. Whereas the UV/O$_3$ process requires both an energy-intensive UV lamp and ozone generator, it could have resulted in high operational costs. However, since it can achieve high degradation efficiencies of contaminants [167], the average $E_{EO}$ values are not as high. For photocatalysis, articles presented in this study were lab-scaled reactors, resulting in a wide range of $E_{EO}$ values, ranging from 0.000038 kWh m$^{-3}$ for a simple treatment of 1 mg/L of tris-(2-chloroisopropyl) phosphate to treating pesticides-containing wastewater with a COD of 1130 mg/L at 233 kWh m$^{-3}$. A similar AOP treatment setup

might yield different $E_{EO}$ values based on the degradation performance of the contaminant. For example, $UV/H_2O_2$ in the degradation for domestic wastewater treatment (0.22 kWh m$^{-3}$) [168] is much lower than synthetic pharmaceutical wastewater treatment (322 kWh m$^{-3}$) [169]. The photocatalyst presented relatively higher average $E_{EO}$ values. However, the higher values presented are due to the use of blacklight UV lamps [135]. Excluding the values computed with the blacklight UV lamps, the photocatalyst has a relatively low average $E_{EO}$ value of 4.77 kWh m$^{-3}$. In the UV/combination list, the $E_{EO}$ values ranged from 0.32395–280 kWh m$^{-3}$, as the combined process may be even more costly as more chemicals or operational costs from the additional equipment associated with the respective combined processes. Higher treatment efficiencies can result in lower treatment time and drive down the $E_{EO}$ as reported by Sgroi et. al. in the treatment of micro-pollutants found in tertiary wastewater effluent [170], whereas UV/Fenton had a relatively high $E_{EO}$ value due to the relatively poorer degradation performance. Solar/Fenton overall had a lower $E_{EO}$ value due to its low energy requirement, yet good degradation performance of CECs.

To circumvent the high cost associated with the operation of UV lamps, there has been growing interest in the use of solar-powered processes with a lower $E_{EO}$ value. Solar/Fenton presents low $E_{EO}$ values across various wastewater treatment applications: The average $E_{EO}$ value of solar/Fenton is also much lower than UV/Fenton at 2.62 and 16.67 kWh m$^{-3}$, respectively. Furthermore, operating the solar/Fenton process at a pilot-scale did not significantly increase the $E_{EO}$ values (4.39 kWh m$^{-3}$), as reported by Expósito et al. [171]. This shows that the solar/Fenton process has potential for scale-up operations and more studies can be done on this aspect for the overall management of CECs. While the computation of $E_{EO}$ might not be a fair comparison when the degradation nature of CECs is so vastly different, it provides a good indication of the potential cost for the degradation of such wastewater. UV and UV/Oxidant processes are also not as effective for hard to degrade compounds such as perfluorooctanoic acid (PFOA), perfluorooctane sulfonate (PFOS), and bisphenol A (BPA), whereas solar/Fenton and UV/catalysis show great potential due to their generally low $E_{EO}$ values. Besides, solar/catalyst also has the potential to be further developed with potentially lower $E_{EO}$ values than UV/catalysis processes. However, due to limited research on solar/catalyst, a reliable $E_{EO}$ value could not be obtained as a comparison.

Based on the literature review, the cost–benefit analysis of each process is summarized in Table 9. Generally, UV/oxidant processes are disadvantageous due to the selective nature of radical generation and low degradation performance of CECs, despite the ease of implementation. $UV/H_2O_2$ has a low molar absorption coefficient of $H_2O$ ($\varepsilon$ = 18.6 M$^{-1}$ cm$^{-1}$ at 254 nm) resulting in the poor generation of $\bullet OH$ radicals [172]. For UV-persulfate and $UV/Cl_2$ processes, the radical species generated are more selective than $\bullet OH$. $S_2O_8^{2-}$ was shown to display higher sensitivity to the DOM composition of water matrix as compared to $UV/H_2O_2$ [173], while $\bullet Cl$ reacts readily with electron-rich contaminants [53]. pH controls are also crucial in the stability of radical species [53].

$UV/O_3$ has a relatively high degradation performance which resulted in a lower-than-expected $E_{EO}$ value. However, $UV/O_3$ is a rather expensive method, as it utilizes both UV lamps and ozone generator(s), which are both energy-intensive [54]. Ozone also has a relatively limited UV absorbance wavelength [110] and pH is usually more effective at pH above 9, which makes it unsuitable for CEC treatment.

For photo-Fenton, heterogeneous Fenton can utilize a broadband of wavelengths of light (180–400 nm). However, with low quantum yield, it results in poor degradation performance [56]. Hence, recent research focuses on the use of Fe (III) ligands (e.g., ethylenediamine-N,N'-disuccinic acid (EEDS), ethylenediaminetetraacetic acid (EDTA), etc.) which have a higher quantum yield due to higher UV absorption [56]. These ligands also have the capabilities of utilizing a broader band of solar energy (180–800 nm) as reported by Soriano–Molina et al. [121].

**Table 9.** Advantages and disadvantages of light-driven AOPs.

| Name of AOP | | Advantages | Drawbacks | References |
|---|---|---|---|---|
| UV/Oxidant | UV/$H_2O_2$ | Cheap and easy to implement | Low degradation performance<br>Control of peroxide dosage needed to meet discharge limits | [51] |
| | UV-persulfate | Fast reaction time<br>Large pH range | Selective degradation at pH below 7, since $SO_4^{\bullet-}$; is the dominant free radical<br>pH adjustment needed | [52] |
| | UV-chlorine | Easy to implement<br>Higher kinetics constants with CECs than HO• | Formation of disinfection by-products<br>Selective degradation of electron-rich moieties compared to unsaturated C-C bonds | [53] |
| UV/Ozone | Microbubble | Fast reaction time<br>Low $E_{EO}$ | High operational and capital cost<br>Works better at elevated pH<br>The limited usable wavelength of light | [54] |
| Photo-Fenton | Heterogeneous | Does not require sludge treatment<br>The large bandwidth of light usage | Slower reaction kinetics | [174] |
| | Homogeneous | Fast reaction time<br>pH range can be extended via the use of chelators like EDDS<br>Able to use solar energy | Large surface area needed for solar processes | [70,175] |
| Photocatalysis | $TiO_2$ | Able to use solar energy with morphology modifications | High energy consumption is needed for the activation<br>Requires high energy UV wavelength<br>Conventional photocatalyst has issues with the reuse of the catalyst | [57,61,176] |
| | ZnO | Able to utilize sunlight without any modifications<br>Cheaper catalyst compared to $TiO_2$ | High energy consumption is needed for the activation<br>Conventional photocatalyst has issues with the reuse of the catalyst<br>Degradation performance was not as good as the titanium-based catalyst | [59] |
| | Alternative catalyst | Able to use solar energy | Relatively new and requires further study of efficiency<br>Suffers potential issues with conventional $TiO_2$ and ZnO issues like the reuse and separation of catalyst | [137,138,141] |

Photocatalysts are generally hampered by the leaching of catalysts as well as the reusability of the catalyst. To combat these issues, researchers doped ferromagnetic materials to $TiO_2$ for easy retrieval of catalysts [57,61,114]. The use of solar irradiation is also currently being explored, with modification of $TiO_2$ based catalyst, to allow solar light to be used as an irradiation source [57]. While $E_{EO}$ for solar-based $TiO_2$ has not to be computed, due to a general lack of study in this sector, this is a potential area for development, whereas zinc-based catalyst is generally not as effective due to inefficiency in the utilization of band gaps in light irradiation. An alternative new catalyst, such as $WO_3$, and carbon-based catalysts suffer similar separation issues and efficiency, although they show potential to use sunlight/visible light as a source of irradiation.

## 4. Conclusions

In recent years there has been growing emphasis and efforts made for the management of CECs. Due to the rapid industry changes and growing concerns about the impact of CECs, the management of CECs is particularly challenging. CECs were conventionally classified based on their polarity and hydrophobicity ($K_{ow}$). However, polarity and $K_{ow}$ of CECs are dependent on their molecular structures and hence the removal of CECs by light-driven processes could be better estimated from their molecular structures. A commonality among other unlisted CECs with reported CECs would serve as a basis for its degradation performance, which is an attractive means to manage newer and insufficiently studied CECs. The occurrence and fate of CECs also suggest that higher concentrations are detected at the effluents of treatment plants. Various point sources were found to concentrate at the effluent of water treatment facilities, due to their recalcitrance to biodegradation. Photo-driven processes for this application are particularly attractive as the effluent characteristics are particularly suitable for photo processes. Effluent is generally low in TSS and NOM after the pre-treatment step. This review then proposed a two-pronged approach in the overall management of CECs, which includes the photo remediation and detection of such contaminants. Key factors that affect light-driven processes were discussed: pH, water matrix, oxidant dosage, catalyst dosage, UV absorbance, and degradation mechanisms.

UV/oxidant processes are generally more selective in the degradation of CECs. As found in the review, UV/$H_2O_2$ has low values of $H_2O_2$ molar absorption coefficient at 254 nm. Photo oxidant processes use persulfate and its derivatives were found to be better at CECs degradation. However, this process requires pH adjustment and the radicals generated to have a lower charge compared to other radicals formed. Thus, the lower oxidative power of the generated radicals leads to selective CEC degradation. UV/chlorine process has a higher reported degradation performance than UV/$H_2O_2$. UV/chlorine uses chemicals that are common for pipeline disinfection for wastewater treatment plant effluent, and hence little modification of the treatment process is needed for immediate application. However, studies noted the formation of undesirable DBPs, and more studies need to be done to limit the formation of such products. Further study is needed on the mechanism and modeling of UV/chlorine to help limit the formation DBPs, for large-scale application of such a process. The computed $E_{EO}$ values of the UV/oxidant process are also the second-lowest next to the solar/Fenton process, which makes this technology particularly attractive.

The UV/ozone process has been demonstrated to have great performance in the degradation of CECs in smaller-scaled reactors and small-scaled pilot studies. However, the high cost of operation and high capital cost of ozone reactors makes this technology not as attractive as other photo technology used in this review. Computed $E_{EO}$ values are the second-highest of all the photo-based technology discussed.

Photo-Fenton has been applied to various large-scale and pilot plant applications, with varying degrees of success. CECs with functional groups that form intermediate compounds like hydroxylated derivatives may impede the performance of Photo-Fenton based degradation. Heterogenous photo-Fenton is shown to be not as popular due to lower reaction kinetics and issues with the catalyst recovery. However, the recent development of solar/Fenton shows promising degradation performance and cost-effectiveness. Computed $E_{EO}$ values showed that solar/Fenton has the lowest $E_{EO}$ values compared to the photo-based treatment of CECs in this review. Coupled with the higher UV wavelength bandwidth of solar/Fenton makes this technology attractive for upscaling, with a key drawback of a high footprint. Ongoing research has been done to reduce the footprint of Solar/Fenton systems. Innovative designs could be done to ensure a larger surface area with a low footprint could be achieved, such as the use of coiled tube reactors or raceway pond designs. Another avenue for further development is the use of high quantum yield ligands which have the ability to utilizing a broader band of solar energy (180–800 nm).

Photocatalysis has a generally lower CECs degradation performance as compared to other photo-based treatment technology, with $TiO_2$ based catalyst having a relatively higher

degradation performance. The computed $E_{EO}$ value was the second-highest next to the UV/ozone process. The lower degradation yield is likely due to the operation in a liquid-solid system, resulting in poor mass transfer. As such, many researchers attempted to circumvent this issue by using the nano-sized catalyst. However, the recovery of this nano-catalyst after usage has been an ongoing issue bottlenecking the large-scale application of these processes. Ongoing research has been done to either attach catalysts onto surfaces of reactors or activated carbon to achieve ease to solid-liquid separation. Alternatively, other researchers used magnetic separation as a means of solid–liquid separation by doping $TiO_2$ onto magnetic materials. Due to the highly specific bandgaps needed to excite the catalyst, the general wavelength needed for photocatalysis is of the higher energy bandwidth of 245–400 nm. Hence, ongoing research has been done to decrease the bandgap between the energy states of the catalyst. Some of the ways utilized were also explored in this review, such as the use of doping other transition metals which were also reported to be able to use solar energy as a light irradiation source. Graphitic carbon nitrides (g-$C_3N_4$) were demonstrated to enhance photocatalyst and photo-Fenton as a composite material. Ongoing investigation has also been done on newer photocatalysts like $WO_3$ and other carbon-based catalysts to further expand the irradiation bandwidth of light used. Further studies on this aspect could be done for the greater application of such technology. Due to its novelty, limited demonstration of degradation performance was done on a larger scale and in a real wastewater matrix. Another consideration is the cost of the production of these newer materials and the potential toxicity of such materials.

Lastly, light-driven monitoring and detection processes were also highlighted as a potential cost-effective replacement for conventional HPLC or GC detection. The current online monitoring of CEC is based on the removal of surrogate parameters in a specific water treatment process. Although light-driven monitoring and detection still need further research, it is undoubtedly a promising technology and could be served as an early-warning system for CECs contamination.

**Supplementary Materials:** The following are available online at https://www.mdpi.com/article/10.3390/w13172340/s1, Supplementary S1 An illustrated example of similar chemical structure of CECs with similar photodegradation performance; Supplementary S2 Bond dissociative energy and their corresponding threshold wavelength; Supplementary S3 Standard Reduction Potentials in Aqueous Medium for degradation of organic compounds; Supplementary S4 Cost Comparison of various light driven AOPs discussed in this review.

**Author Contributions:** B.C.Y.L.: Conceptualization, investigation, writing—original draft preparation. F.Y.L.: Overall review, editing, investigation, visualization. W.H.L.: Investigation, review, and editing; S.L.O.: Fund acquisition and project management, J.H.: Fund acquisition and project management, overall review. All authors have read and agreed to the published version of the manuscript.

**Funding:** This research was funded by Sembcorp Corporate Laboratory under the "Achieving low COD in industrial treated effluent through combination of AOPs and biological processes" research project, grant number WBS: R-261-513-005-281.

**Institutional Review Board Statement:** Not applicable.

**Informed Consent Statement:** Not applicable.

**Data Availability Statement:** The data presented in this study is available on request from the corresponding author.

**Acknowledgments:** This research is supported by the National Research Foundation Singapore, Sembcorp Corporate Laboratory and the National University of Singapore.

**Conflicts of Interest:** The authors declare no conflict of interest.

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
