# Peer review of "Emerging Contaminants: An Overview of Recent Trends for Their Treatment and Management Using Light-Driven Processes"

_water, doi:10.3390/w13172340_

Round 1

Reviewer 1 Report

The manuscript has been corrected and, in my opinion, it can be accepted for publication in the current form.

Reviewer 2 Report

Review Report

MS Title:        Emerging contaminants: an overview of the recent trends for light-driven processes for their treatment and management

MS ID:            water-1346156

Journal:          Water

General comment

Lee and co-authors did an excellent review of emerging scientific progress for remediation of emerging organic pollutants using light-related processes. The review is timely and critical in many instances. I have suggested minor adjustments detailed as specific comments below to improve the quality of the report.

Specific Comments

  1. Referencing and citation format are not aligned with the journal requirements.
  2. Line 32-33. Although one may partially agree with the authors about the unregulated status of the majority of CECs, however, there are some legislative guidelines recently in place in some countries. Please refer to https://doi.org/10.1016/j.jece.2021.105966 and revise the statement accordingly. One way to correct the sentence is to insert “in many countries” after the word “unregulated”. The authors have even acknowledged the fact about regulation in Line 42.
  3. Line 39: Replace “they are costly” with “prohibitive cost”.
  4. Line 55-56: Confusing sentence. Check and revise.
  5. The content of Section 1.1 does not really address the section heading. For concise definitions of CECs, the authors should please refer to https://doi.org/10.1007/s42250-019-00079-6. The section should be revised accordingly.
  6. Table 1 should be revised to incorporate relevant locations/regions/countries to reflect global trend. Please refer to the following reports from Africa which is missing in the Table. https://doi.org/10.1016/j.watres.2014.08.002, https://doi.org/10.1016/j.scitotenv.2020.142177, https://doi.org/10.1007/s00128-015-1639-9.
  7. In Figure 1, change “Affecting Factors” to “Controlling Factors”.
  8. Line 160. Correct to “It should also be noted” ….
  9. Table 3. Wrong symbol for oxygen atom in H2O2. See the 4th row of the table in page 10.
  10. Line 196, page 10. What is HRT? HRT is mentioned for the first time here. All abbreviations should be defined in full at first mention.
  11. Table 4. Check and correct the unit for energy consumption in the 4th row of Table 4 in page 12.
  12. Check and correct the formula of H2O2 in page 21, 4th row of Table 6.
  13. Line 273. Confusing sentience. Check and correct. Punctuation may be required.
  14. Line 319. “photocatalysis” not “photocatalytic”
  15. Line 326. Check the spelling of “amide”.
  16. Check the citation format for Juang et al. 2018. The reference is also not listed in the reference list.
  17. The authors should provide some relevant citations for the examples of new photocatalysts mentioned. The authors should consider citing the recent paper in Water (https://doi.org/10.3390/w13141918) for C-based photocatalyst for light-driven degradation of tetracycline.
  18. Line 399. Check the spelling of “physico-chemical”.
  19. Line 404. Check the superscript for R2.
